# Human microcephaly protein RTTN interacts with STIL and is required to build full-length centrioles

Hsin-Yi Chen[1,2], Chien-Ting Wu[2,3], Chieh-Ju C. Tang[2], Yi-Nan Lin[2], Won-Jing Wang[4] & Tang K. Tang [1,2,3]

Mutations in many centriolar protein-encoding genes cause primary microcephaly. Using super-resolution and electron microscopy, we find that the human microcephaly protein, RTTN, is recruited to the proximal end of the procentriole at early S phase, and is located at the inner luminal walls of centrioles. Further studies demonstrate that RTTN directly interacts with STIL and acts downstream of STIL-mediated centriole assembly. CRISPR/Cas9-mediated *RTTN* gene knockout in p53-deficient cells induce amplification of primitive procentriole bodies that lack the distal-half centriolar proteins, POC5 and POC1B. Additional analyses show that RTTN serves as an upstream effector of CEP295, which mediates the loading of POC1B and POC5 to the distal-half centrioles. Interestingly, the naturally occurring microcephaly-associated mutant, RTTN (A578P), shows a low affinity for STIL binding and blocks centriole assembly. These findings reveal that RTTN contributes to building full-length centrioles and illuminate the molecular mechanism through which the RTTN (A578P) mutation causes primary microcephaly.

[1] Graduate Institution of Life Sciences, National Defense Medical Center, Taipei, Taiwan. [2] Institute of Biomedical Sciences, Academia Sinica, Taipei, Taiwan. [3] Taiwan International Graduate Program in Interdisciplinary Neuroscience, National Yang-Ming University and Academia Sinica, Taipei, Taiwan. [4] Institute of Biochemistry and Molecular Biology, College of Life Sciences, National Yang-Ming University, Taipei, Taiwan. Correspondence and requests for materials should be addressed to T.K.T. (email: tktang@ibms.sinica.edu.tw)

The centriole is a conserved microtubule-based organelle that is an essential component of centrosomes, cilia, and flagella. The formation of a new centriole adjacent to a pre-existing centriole is a highly ordered process that can be broadly divided into the stages of initiation, elongation, and maturation[1–3]. In vertebrate cells, centriole duplication occurs during the late G1-S phase. A new daughter centriole, termed a procentriole, starts to grow orthogonally from the proximal end of a pre-existing centriole, elongates through the S and G2 phases, and reaches its full length in the early phase of mitosis. The formed centriole becomes a fully mature mother centriole in the following G1 phase, when it acquires two sets of subdistal/distal appendages.

In mammalian cells, several evolutionarily conserved proteins have been shown to participate in different stages of the centriole duplication process. At the initiation stage, PLK4, STIL, SAS-6, and CEP135 are the key elements responsible for assembling the cartwheel[4–10]. STIL recruits CPAP to the outer region of the cartwheel[8], where it assembles 9-triplet centriolar microtubules[11–13]. During centriole elongation, CEP120, SPICE, and centrobin have been reported to regulate centriole elongation via direct interactions with tubulin and one another[14–16], while CEP295, POC5, and POC1B are required to build the distal portion of the centriole[17–20].

Rotatin (RTTN) is a centrosome-associated protein that is evolutionarily conserved in many organisms. The *RTTN* gene was initially identified in a homozygous mutant mouse that shows defects in axial rotation and left–right specification[21]. Ana3, which is the *Drosophila* homolog of RTTN, shares only 19% amino acid sequence identity with human RTTN. It is reportedly needed to ensure the structural integrity of centrioles and basal bodies, while being dispensable for centriole duplication[22]. *RTTN* mutations were previously identified in human patients with polymicrogyria, which is a cilia-defect-associated malformation of the developing cerebral cortex[23]. Recently, homozygous mutations in the *RTTN* gene were reported to cause primary microcephaly (MCPH) and primordial dwarfism in humans[24]. However, the roles of RTTN in centriole function and ciliogenesis remain largely unknown.

In this study, we uncover for the first time the function and action mechanism of RTTN in the centriole biogenesis of human cells. We show that RTTN is a STIL-interacting protein that acts downstream of STIL-mediated centriole duplication. Our results demonstrate that RTTN is not essential for initial centriole assembly; instead, it is required for assembly of full-length centrioles. Moreover, we report that the MCPH-associated RTTN (A578P) mutant exhibits a decreased affinity for STIL and inhibits centriole duplication. This suggests that the STIL-RTTN interaction is critical for proper centriole biogenesis, and that dysfunction of this interaction may cause MCPH in humans.

## Results

**RTTN loss impairs centriole elongation and induces PPBs.** Although Ana3 was previously reported to be dispensable for centriole duplication in *Drosophila* cells[22], mutations in the *RTTN* gene (the human homolog of Ana3) cause MCPH[24]. To investigate the centrosomal role of RTTN in human cells, we depleted RTTN from U2OS cells (p53 wild-type) using specific siRNA duplexes (siRTTN#1, #2, and #3), as presented in Fig. 1a. Our western blotting and immunofluorescence staining results showed that all three siRNAs substantially, but not completely, inhibited RTTN expression in U2OS cells (Fig. 1b, c). The number of centrioles (centrin-positive) was significantly reduced in all three siRTTN-treated U2OS cells (<4; Fig. 1c, d). Interestingly, the early-S-nascent centrioles contained STIL[8] and SAS-6[6] (two known early-born centriolar proteins, Fig. 1e, f), but

the G2-nascent centrioles lacked POC5[17] (a later-born protein, arrow, Fig. 1h) in siRTTN#1-treated U2OS cells. Consistent with this finding, depletion of RTTN (siRTTN#1, hereafter referred to as siRTTN) did not interfere with the targeting of early-born proteins (SAS-6, CP110, and centrin) to the early-S-nascent centrioles (Supplementary Fig. 1a–d), but it did suppress the recruitment of later-born centriolar proteins (POC1B and POC5) to the G2-nascent centrioles (Supplementary Fig. 1f, g) in U2OS-based PLK4-myc-inducible cells. These cells were selected because the localization of target proteins on amplified newborn centrioles surrounding the pre-existing centrioles could be clearly visualized upon PLK4 induction[5].

To further examine the centrosomal role of RTTN in centriole biogenesis, we generated stable RPE1 cell lines in which *RTTN* and *p53* genes were mutated ($RTTN^{-/-}$; $p53^{-/-}$) using the CRISPR/Cas9-mediated gene editing system. The latter mutation was introduced because acentriolar cells are not viable in the presence of p53[25]. Two independent *RTTN*-knockout cell lines (#1 and #2) were obtained and verified by genomic DNA sequencing (Supplementary Fig. 2a, b). Western blotting (Supplementary Fig. 3a) and immunofluorescence analysis (Supplementary Fig. 3b-iv) of $RTTN^{-/-}$; $p53^{-/-}$ #1 cells (hereafter referred to $RTTN^{-/-}$; $p53^{-/-}$) revealed complete loss of RTTN signals. Strikingly, no "typical" centrioles were detected in $RTTN^{-/-}$; $p53^{-/-}$ cells. Instead, we detected de novo formation of numerous primitive procentriole bodies (PPBs) that were positive for centrin (a centriole marker) (Supplementary Fig. 3b-iv, c). These PPBs contained the early-born centriolar proteins, STIL (Supplementary Fig. 3d), SAS-6 (Supplementary Fig. 3e), and CPAP (Supplementary Fig. 3f), but lacked the later-born centriolar proteins, POC1B (Supplementary Fig. 3g) and POC5 (Supplementary Fig. 3h). Similar characteristics were observed in the second line of $RTTN^{-/-}$; $p53^{-/-}$ cells (#2). These results suggest that the loss of RTTN does not affect initial procentriole assembly; instead, it appears to impair the later stage of centriole elongation.

Interestingly, the formation of PPBs seem to be cell cycle regulated. Tracing with GFP-CP110 (an early-born centriolar protein) and mCherry-H2B (histone H2B, a chromosome marker; Supplementary Fig. 4a and Supplementary Movie 1) or GFP-CP110 and mCherry-STIL (early-born centriolar proteins; Supplementary Fig. 4b and Supplementary Movie 2) revealed that the PPBs appeared at early S phase and gradually disassembles when cells entered mitosis. Previous reports showed that STIL[8] and SAS-6[6] are cell cycle-regulated proteins that are degraded in late mitosis. It is thus possible that the PPBs could not be formed in the absence of STIL and SAS-6 during mitosis. Furthermore, depletion of PLK4 (a key enzyme that controls the initial steps of centriole duplication[4]) or addition of centrinone B[26] (a PLK4 inhibitor) could block the formation of PPBs that lack of STIL, SAS-6, and CPAP in $RTTN^{-/-}$; $p53^{-/-}$ cells (Supplementary Fig. 5a–c). Importantly, the phenotype of numerous PPBs (>4) could be effectively rescued and converted to a normal phenotype (2 or 4 centrioles/cell) by exogenous expression of wild-type RTTN-GFP (Supplementary Fig. 3b-iii, c). The rescued centrioles contained POC5 (Supplementary Fig. 5e) and CEP164 (a distal appendage protein, Supplementary Fig. 5f), and exhibited the proper nine-triplet microtubule arrangement (Supplementary Fig. 5g). Collectively, our results suggest that instead of being non-functional improperly organized protein aggregates, PPBs are more likely to represent primitive procentrioles that possess the ability to form normal centrioles. However, the possibility of PPBs being as protein aggregates cannot be ruled out.

Unexpectedly, no de novo PPBs were found in siRTTN-treated U2OS cells (Fig. 1). This might reflect the presence of a small amount of leftover RTTN, which could block the formation of

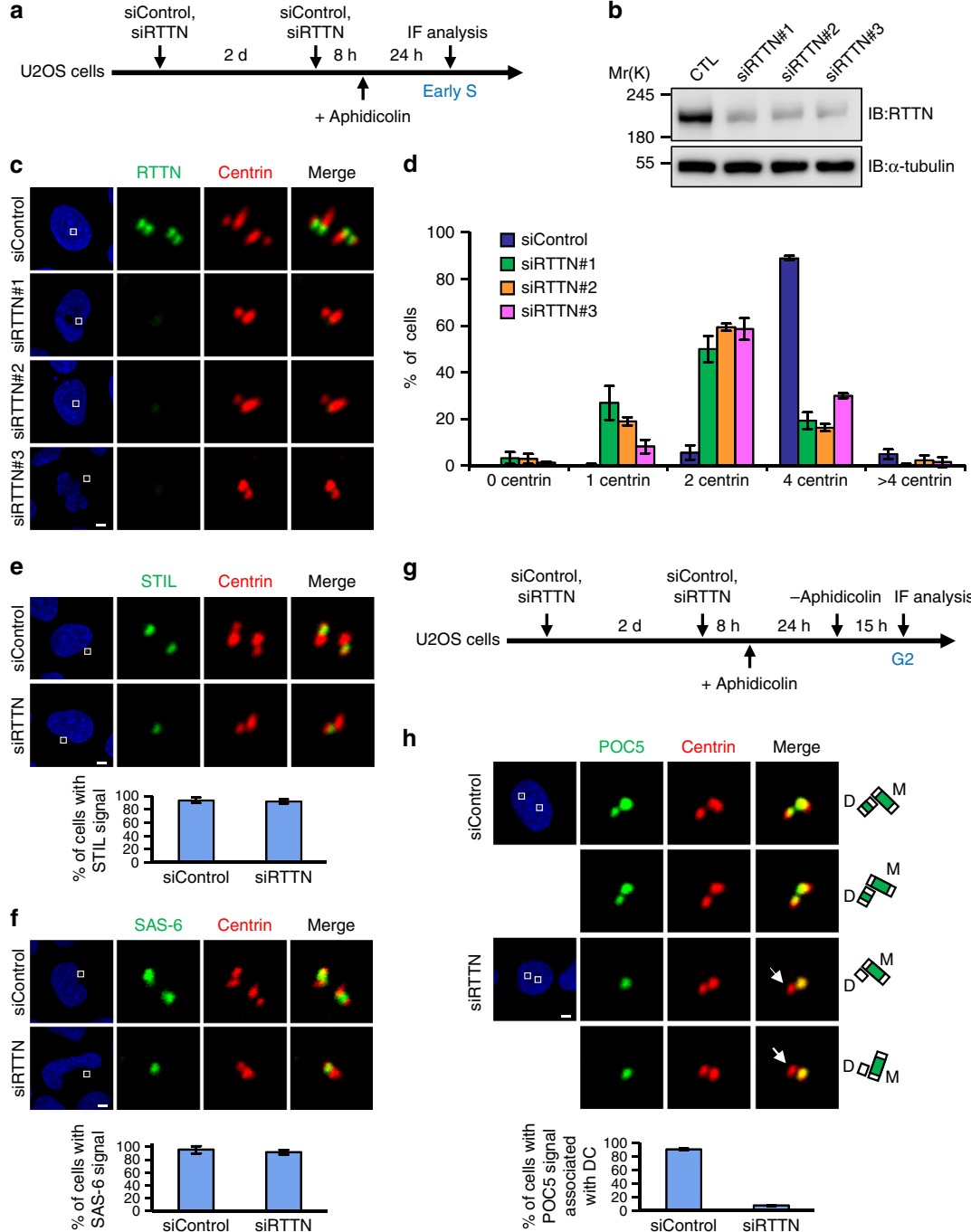

**Fig. 1** RTTN is required for normal centriole duplication. **a–c** U2OS cells were treated with siControl or siRTTN (#1, #2, #3) and synchronized at early S phase, as described in **a**. The cells were analyzed by immunoblotting **b** or immunofluorescence confocal microscopy **c** using the indicated antibodies. **d** *Histogram* illustrating the percentages of cells exhibiting centrin signals. *Error bars* represent the mean ± s.d. from three independent experiments ($n = 100$/experiment). **e, f** U2OS cells were treated with siControl or siRTTN as described in **a**, and then stained with the indicated antibodies. *Histogram* illustrating the percentages of cells exhibiting STIL **e** or SAS-6 **f** signals. *Error bars* represent the mean ± s.d. from three independent experiments ($n = 100$/experiment). **g, h** U2OS cells were treated with siControl or siRTTN, enriched at G2 phase as described in **g**, and stained with the indicated antibodies. *Histogram* illustrating the percentages of cells with a POC5 signal associated with daughter centriole (DC). *Error bars* represent the mean ± s.d. from three independent experiments ($n = 100$/experiment). *Scale bar*, 5 μm

PPBs in p53 wild-type cells, and/or it could indicate that p53 negatively regulates PPBs formation via a yet-unknown mechanism. Taken together, our results suggest that: (1) RTTN is not essential for initial procentriole assembly; (2) RTTN is required for the loading of the later-born centriolar proteins, POC1B and POC5, to the distal-portion centrioles; and (3) loss of p53 does not appear to affect centriole elongation.

**Depletion of RTTN produces shorter centrioles**. As the procentriole is known to elongate during S-G2 phases[1], we carefully examined the elongation of procentrioles labeled with antibodies against SAS-6 and CP110 in PLK4-myc-inducible cells during S-G2 phases. The PLK4-myc-inducible cells were treated with siControl or siRTTN (Fig. 2a–d), exposed to aphidicolin, and then released at different time points, as described in Fig. 2e. The

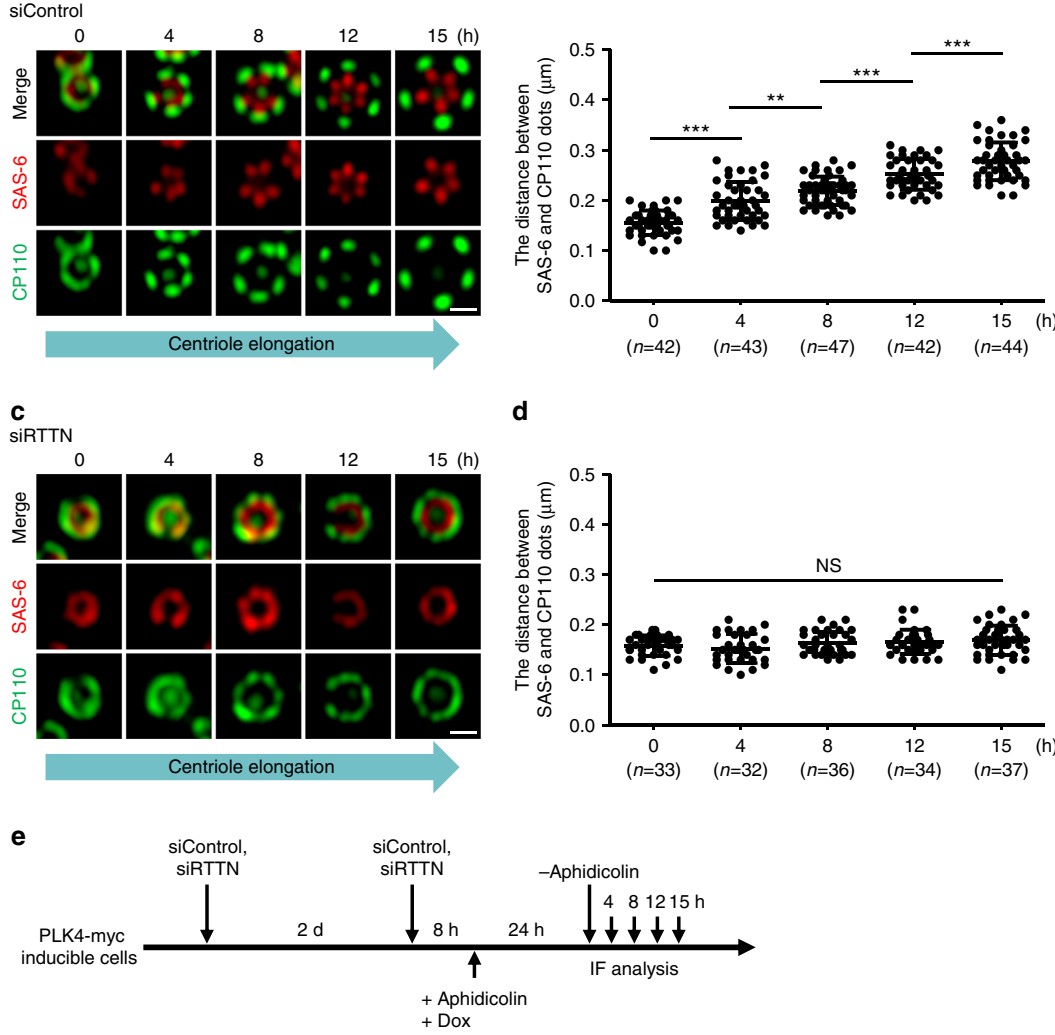

**Fig. 2** Depletion of RTTN perturbs centriole elongation. PLK4-myc-inducible cells were treated with siControl **a**, **b** or siRTTN **c**, **d** as shown in **e**, and analyzed by confocal fluorescence microscopy using the indicated antibodies. The procentriole length was measured as the distance between the fluorescent peak intensity of SAS-6 (*red*) and CP110 (*green*) in siControl- **a**, **b** and siRTTN-treated cells **c**, **d**. *Error bars* represent the mean ± s.d.; **P < 0.01; ***P < 0.001; NS, not significant (two-tailed *t*-test). *Scale bar*, 0.5 μm

cells were then immunostained with the indicated antibodies. To estimate the procentriole length, the distance between SAS-6 (a proximal end protein) and CP110 (a distal-end protein) signals was quantified in PLK4-induced centrioles (Fig. 2b, d). At the early stages of centriole elongation (0 h, early S), the newly formed CP110- and SAS-6-labeled procentrioles were observed as ring-like structures. As the cell cycle proceeds, these ring-like structures gradually elongated (as measured by the distance between SAS-6 and CP110 signals) to became a clear rosette pattern (15 h) wherein each petal may represent an elongated centriole. In contrast, depletion of RTTN significantly blocks this procentriole elongation, often producing a smear of ring-like structures in PLK4-myc-inducible cells (Fig. 2c, d).

To further elucidate the potential role of RTTN in regulating centriole elongation, RTTN-depleted (siRTTN) or control (siControl) U2OS cells were synchronized at G2 phase. In such cells, the duplicated centrosomes could be clearly distinguished from one another. The cells were then immunostained with antibodies against CEP162 (a known centriole distal-end marker) and acetylated tubulin (acTub, a stabilized centriole marker; see Fig. 3a). The distance between the two CEP162 dots associated with a given pair of centrioles was measured as described

by Azimzadeh et al.[17] Our results showed that the mean distance between two CEP162 dots in siRTTN-treated cells (0.58 ± 0.08 μm) was significantly shorter than that in siControl cells (0.81 ± 0.08 μm; Fig. 3b, c). Consistent with this finding, RTTN depletion also produced multiple shortened G2 centrioles in PLK4-myc-inducible cells, as evidenced by measuring the length of procentrioles (the mean distance between dots of CEP162 and SAS-6; Fig. 3d, e). To more precisely assess the centriole defects in RTTN-depleted cells, we treated PLK4-myc-inducible cells with siRTTN, synchronized the cells at G2 phase, and performed electron microscopy. As shown in Fig. 3f, g, the average procentriole length in RTTN-depleted cells (99.86 ± 18.81 nm, n = 23, P < 0.0001) was significantly shorter than that of siControl cells (259.50 ± 29.52 nm, n = 26). Together, these findings provide further evidence that RTTN plays a role in centriole elongation.

Our group and others previously showed that overexpression of CPAP[11–13] or CEP120[14, 15] could induce the assembly of overly long centrioles (>0.5 μm), whereas depletion of RTTN leads to the production of shorter centrioles (Figs. 2 and 3). To examine whether RTTN is required for CPAP- or CEP120-mediated centriole elongation, we depleted RTTN from CPAP- or

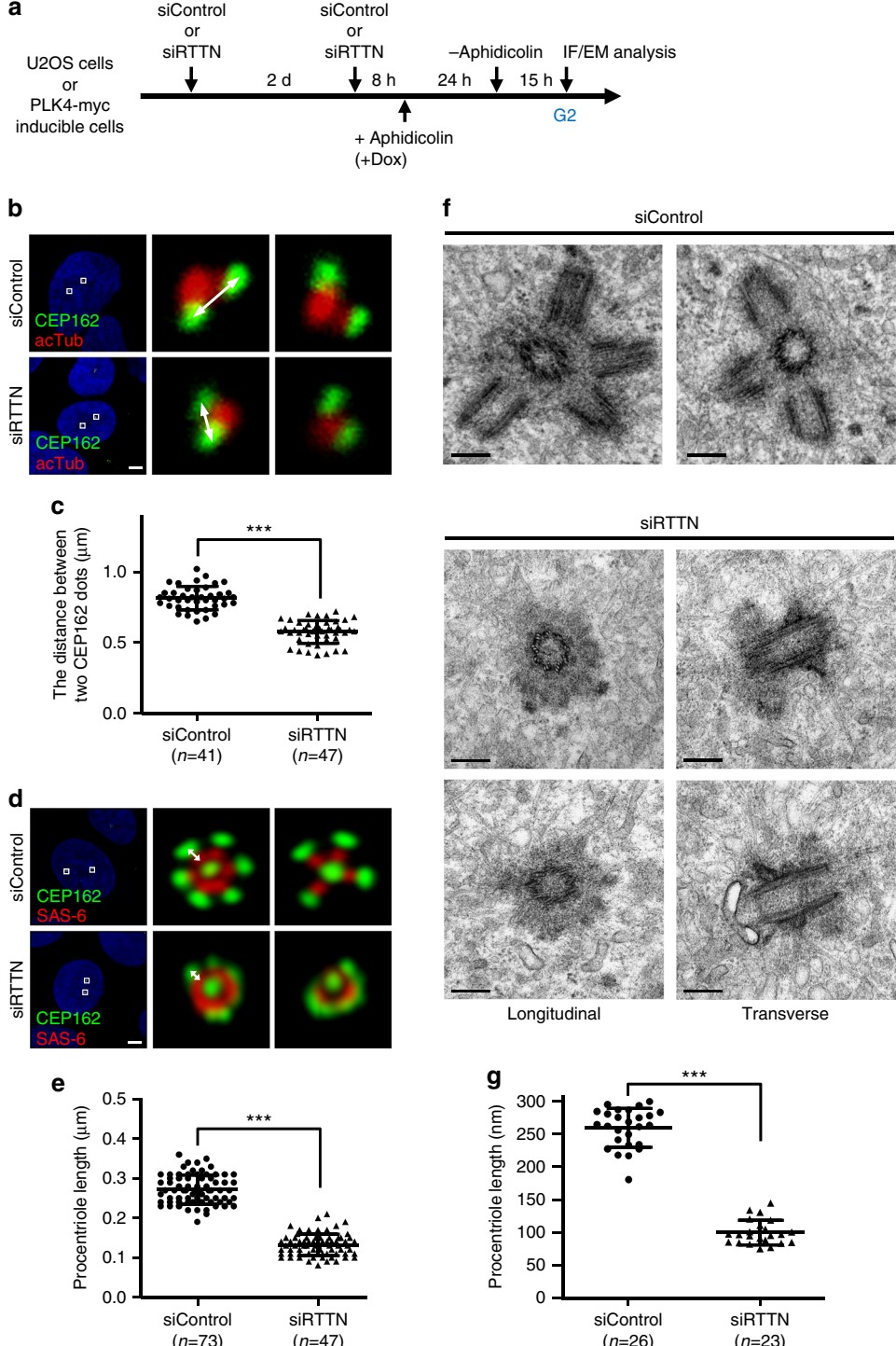

**Fig. 3** Depletion of RTTN produces shorter centrioles. **a**, **b** U2OS cells were treated with siControl or siRTTN as shown in **a**, and analyzed by confocal fluorescence microscopy using the indicated antibodies. **c** *Histogram* illustrating the distance between the CEP162-positive dots associated with a given pair of orthogonally oriented centrioles. *Error bars* represent the mean ± s.d.; ***P < 0.001 (two-tailed *t*-test). **d**, **f** PLK4-myc doxycycline (Dox) inducible cells were treated with siControl or siRTTN as shown in **a**, and then analyzed by confocal fluorescence microscopy using the indicated antibodies **d**, or by electron microscopy **f**. **e**, **g** *Histogram* illustrating the length of procentrioles in PLK4-myc-inducible cells, as analyzed by confocal microscopy **e** or electron microscopy **g**. *Error bars* represent the mean ± s.d.; ***P < 0.001 (two-tailed *t*-test). The procentriole length in b/c was measured as the distance between two CEP162 dots as described by Azimzadeh et al.[17] The procentriole length in d/e was measured as the distance between the fluorescent peak intensity of SAS-6 (*red*) and CEP162 (*green*) in siControl and siRTTN-treated cells. *Scale bar*, 5 μm in **b** and **d**; *Scale bar*, 200 nm in **f**

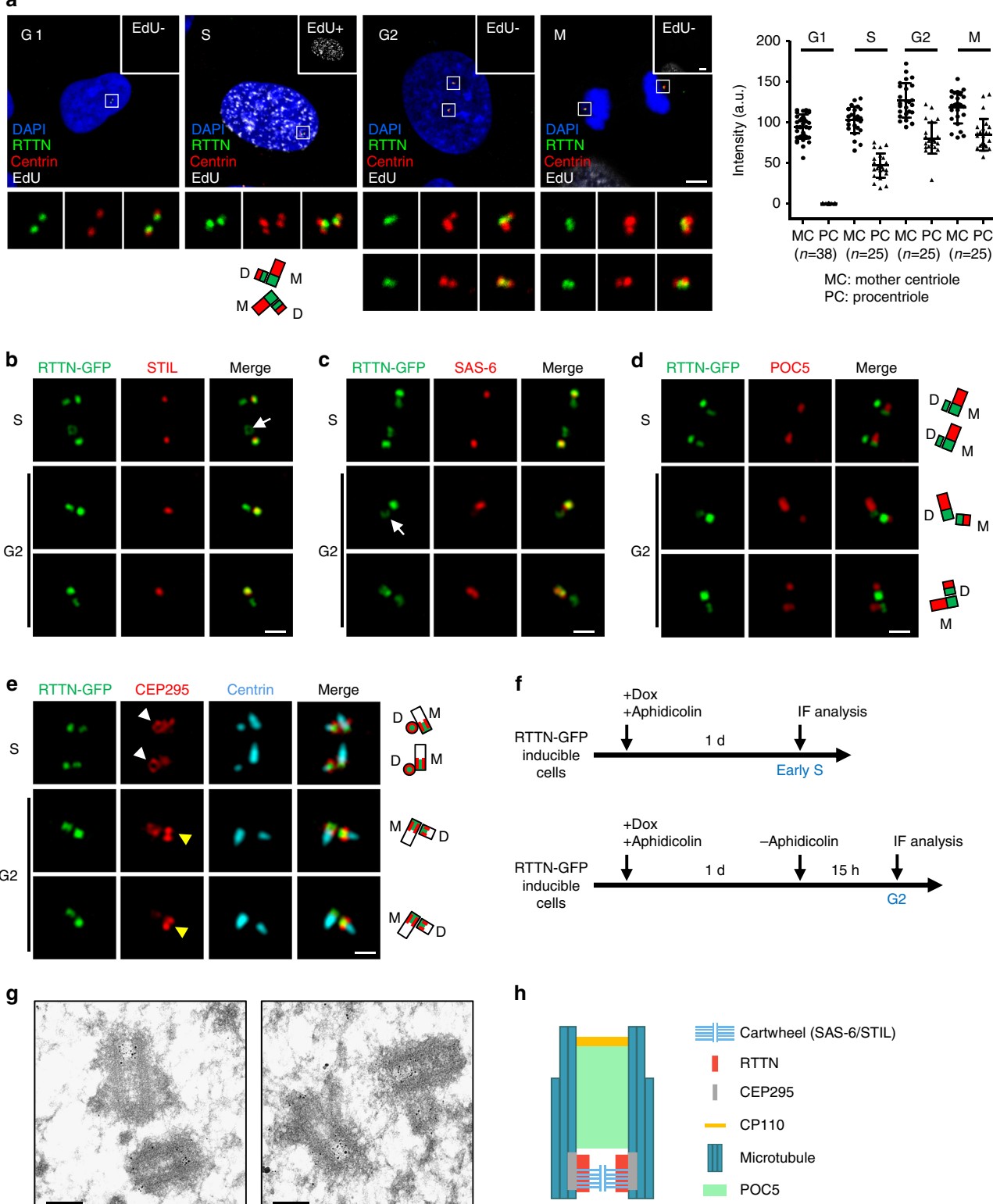

**Fig. 4** Super-resolution (3D-SIM) and immunogold electron microscopic analysis of the localization of RTTN during centriole biogenesis. **a** The subcellular localization of RTTN during the cell cycle. U2OS cells at different cell cycle stages were subjected to EdU labeling (*white*) and immunofluorescence staining using antibodies against RTTN (*green*) and centrin (*red*). EdU-labeled nuclei (EdU+) indicate cells at S phase. DNA was counterstained with DAPI (*blue*). The fluorescence intensity of RTTN at centrioles was quantified by ZEN software. **b**–**f** RTTN-GFP doxycycline (Dox)-inducible cells were treated as described in **f**, and stained with antibodies against STIL **b**, SAS-6 **c**, POC5 **d**, CEP295 **e**, and centrin **e**. The fluorescence images were analyzed by 3D-SIM. M, mother centriole; D, daughter centriole. **g** Immunogold EM analysis of RTTN localization. **h** A *schematic* showing the localization of RTTN relative to those of other centriolar proteins at a centriole. *Scale bar*, 5 µm in **a**; *Scale bar*, 0.5 µm in **b**–**e**; *Scale bar*, 200 nm in **g**

CEP120-inducible cells (Supplementary Fig. 6g) and stained the cells with antibodies against CEP164 (a mother centriole marker) and acetylated tubulin (acTub). Our results showed that the lengths of both mother centrioles (MC) and daughter centrioles (DC) were significantly shorter in RTTN-depleted CPAP-inducible cells compared to siControl cells (Supplementary Fig. 6a–c). Interestingly, most of the RTTN-depleted cells contained a single mature parent centriole (~49%; Supplementary Fig. 6c), whereas the remaining cells contained either one longer CEP164-positive centriole (~12%) or two short centrioles (~37%).

Thus, depletion of RTTN severely impaired the CPAP-myc-induced elongation of both parent and newborn centrioles. Similar effects were observed in CEP120-myc-inducible cells (Supplementary Fig. 6d–f). Together, our results suggest that RTTN is required for CPAP-/CEP120-mediated centriole elongation.

**RTTN is localizes to the inner luminal wall of the centriole**. We next examined the subcellular localization of endogenous RTTN during the cell cycle. The intensities of RTTN at the mother

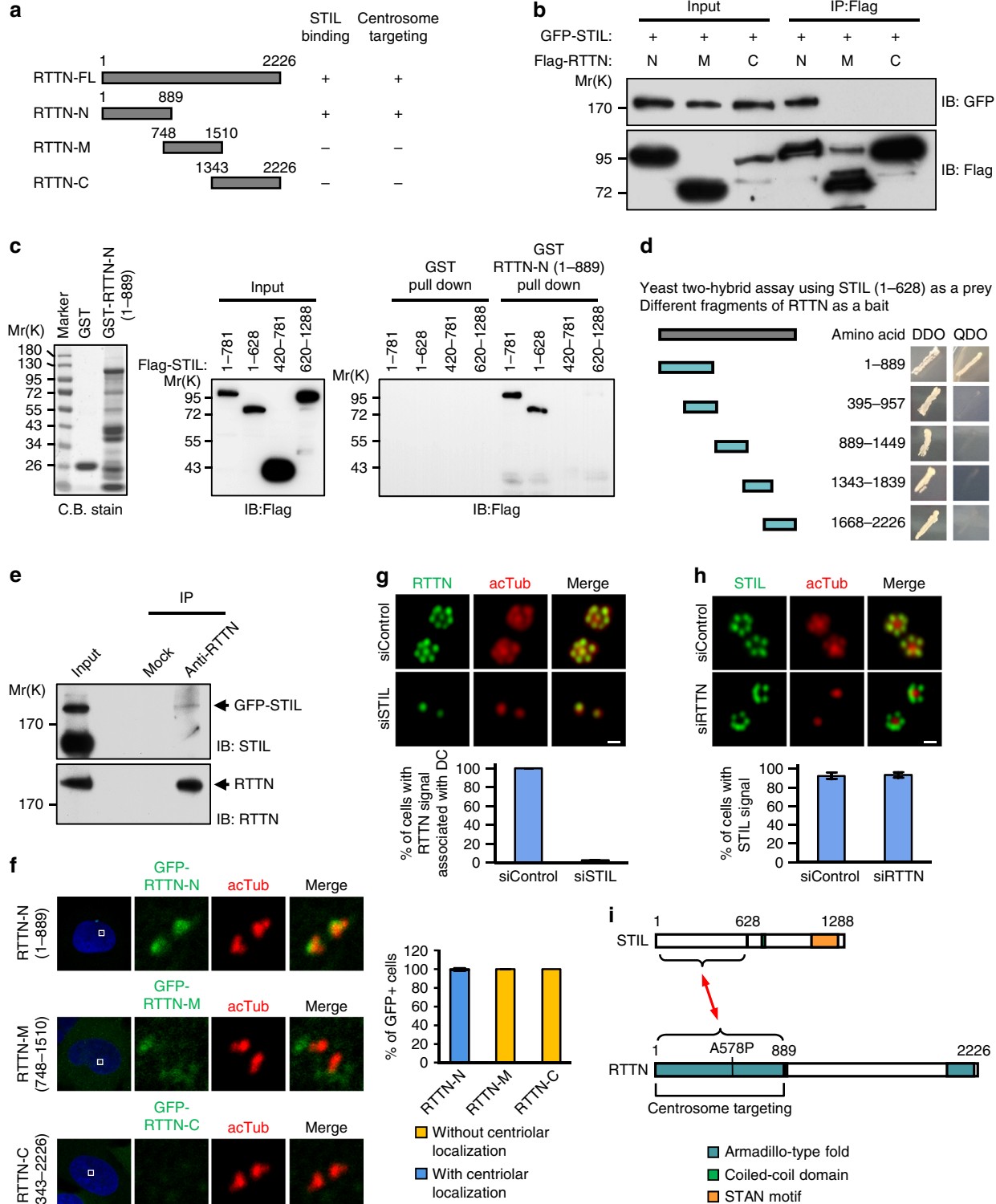

centriole and procentriole were quantified (Fig. 4a, *right panel*). Our immunofluorescence analyses showed that RTTN was associated with both centrioles during G1 phase. Interestingly, the intensity of RTTN at the procentriole, but not the mother centriole, gradually increased during S phase (EdU-positive cells) and finally reached a plateau at G2/M (Fig. 4a, *right panel*). To precisely define the localization of RTTN and examine its spatial and temporal correlation with other centriolar proteins during centriole biogenesis, we performed three-dimensional structured illumination microscopy (3D-SIM) and immunoelectron microscopy. Due to the unavailability of antibodies suitable for examining the co-localization of two different proteins by immunofluorescence staining, we instead generated GFP-tagged RTTN (RTTN-GFP)-inducible cells. We then synchronized the RTTN-GFP cells at early S or G2 phase (Fig. 4f), and used 3D-SIM to analyze the spatial and temporal correlation of RTTN-GFP with other centriolar proteins.

Our results showed that most of the RTTN-GFP signals (*green*) co-localized (*yellow*) with STIL (an inner centriolar lumen protein, *red*; Fig. 4b) and SAS-6 (a protein contained within cartwheels, *red*; Fig. 4c) for early S- and G2-stage centrioles. Interestingly, POC5, a distal-portion centriole protein[17] not found in early S centrioles (Fig. 4d), was clearly detected distal to RTTN-GFP on G2 centrioles (Fig. 4d). We also unexpectedly observed that some RTTN-GFP signals show a ring-like structure on mother centrioles from top–down views (Fig. 4b, c, *arrows*). Collectively, these findings suggest that RTTN could be an inner lumen protein located at the proximal end of centrioles, and that it co-localizes with SAS-6 and STIL.

Consistent with these findings, our immunoelectron microscopy directly demonstrated that RTTN-gold particles were mainly concentrated within the proximal lumen of both mother and daughter centrioles (Fig. 4g). Unfortunately, our 3D-SIM analysis was unable to distinguish the spatial distances between RTTN and STIL or RTTN and SAS-6, due to the limit of its resolution power (100–130 nm)[27]. Furthermore, we previously showed that CEP295 is located at the centriolar microtubule wall that embraces the cartwheel protein, SAS-6[19]. Notably, our top–down views showed that RTTN-GFP was encompassed by CEP295 (*white arrowheads*, Fig. 4e), while the side-view images of CEP295 commonly revealed a two-dot shape with RTTN-GFP in the middle (Fig. 4e, *yellow arrowheads*). Taken together, our results strongly support a spatial model in which RTTN is recruited to the proximal inner luminal wall of centrioles at early S phase, where it co-localizes with STIL and SAS-6 (most likely encircling the SAS-6-containing cartwheel) and is encompassed by CEP295 (Fig. 4h).

**RTTN acts as a downstream effector of STIL.** Given that RTTN co-localized with STIL and SAS-6, and all three proteins are required for centriole duplication, we next used co-immunoprecipitation (co-IP) experiments to examine whether RTTN could form a complex with either STIL or SAS-6. Due to the large size of the RTTN protein (2226 amino acids, (aa)), we sub-divided the full-length RTTN cDNA into three fragments, RTTN-N (the amino-terminal region, aa 1–889), RTTN-M (the middle region, aa 748–1510), and RTTN-C (the carboxy-terminal region, aa 1343–2226), which contained overlapping regions of ~150 amino acids for each fragment (Fig. 5a). A vector encoding GFP-tagged full-length STIL was co-transfected into HEK293T cells along with various Flag-tagged RTTN truncation constructs (N, M, and C; Fig. 5b). Our co-IP analysis showed that GFP-STIL could co-precipitate with RTTN-N but not RTTN-M or RTTN-C, implying that full-length STIL may interact with RTTN-N in vivo (Fig. 5b). We next examined the RTTN-interacting region of STIL using GST-pulldown assays. Various vectors encoding Flag-tagged STIL truncation of STIL were transfected into HEK293T cells, and cell lysates were subjected to pulldown assays (Fig. 5c). Our results showed that GST-RTTN-N (Fig. 5c) could pull down complexes containing STIL (aa 1–628) and STIL (aa 1–781), but not STIL (aa 420–781), or STIL (aa 620–1288), suggesting that the N-terminal region of STIL (aa 1–628) is essential for RTTN inter-action (Fig. 5c). Consistent with this result, our yeast two-hybrid experiments demonstrated that RTTN-N (aa 1–889) directly interacted with STIL (aa 1–628) when the latter fragment was used as a prey (Fig. 5d). Importantly, our co-IP experiments directly demonstrated that endogenous full-length RTTN could form a complex with exogenously expressed GFP-STIL (Fig. 5e). Inter-estingly, the N-terminal domain of RTTN (aa 1–889, RTTN-N) was also required for the localization of RTTN to the centrosome (Fig. 5f), indicating that RTTN-N is required for both centrosome targeting and the STIL interaction (Fig. 5a). Furthermore, we have performed co-IP experiments to examine the possible association of RTTN with other centriolar proteins. We did not find the evidence that RTTN associates with SAS-6, CPAP, CEP135, or CEP295 (Supplementary Fig. 7).

Based on our evidence that RTTN directly interacts with STIL (Fig. 5a–e), we next examined their functional relationship. We depleted STIL from U2OS-based PLK4-myc-inducible cells and examined the centriolar localization of RTTN, and vice versa. As shown in Fig. 5g, the centriolar localization of RTTN at newborn centrioles was strongly diminished upon STIL depletion. In contrast, depletion (Fig. 5h) or complete loss (Supplementary Fig. 3) of RTTN had no apparent effect on the procentriolar recruitment of STIL. Our data thus collectively suggest that RTTN is a downstream effector of STIL, and that RTTN is very likely to be recruited to the proximal end of newborn centrioles via the N-terminal domain of STIL at early S phase. However, since depletion of STIL could completely block the formation of newborn centrioles, further experiments are needed to test the latter hypothesis.

**Fig. 5** RTTN directly interacts with and acts downstream of STIL. **a** Schematic representations of full-length RTTN-FL (1–2226), RTTN-N (1–889), RTTN-M (748–1510) and RTTN-C (1343–2226). **b** Mapping the STIL-interacting domain in RTTN. HEK293T cells were co-transfected with GFP-STIL and various Flag-tagged RTTN constructs, and then analyzed by immunoprecipitation (IP) and subsequent immunoblotting (IB) using the indicated antibodies. **c** GST-pulldown assay. GST and GST-RTTN-N (1–889) recombinant proteins were affinity purified and used to pulldown the indicated proteins from lysates of HEK293T cells that had been transfected with various Flag-STIL constructs. **d** Yeast two-hybrid assay shows a direct interaction between truncated STIL (1–628, prey) and various portions of RTTN (bait). The positive interaction between STIL and RTTN was demonstrated by the growth of mating colonies on QDO plates. **e** Endogenous RTTN and GFP-STIL form a complex in vivo. HEK293T cells were transfected with GFP-STIL. Twenty-four hours after transfection, the cell lysates were immunoprecipitated with anti-RTTN antibodies and immunoblotted with anti-STIL and anti-RTTN antibodies. **f** Mapping the centrosome-targeting region of RTTN. U2OS cells were transiently transfected with various GFP-tagged truncated RTTN constructs. At 24 h post-transfection, the cells were fixed and stained for acetylated tubulin (acTub). **g** RTTN acts downstream of STIL. PLK4-myc-inducible cells were treated with siControl or siSTIL and synchronized at early S phase by aphidicolin treatment. The cells were then subjected to immunofluorescence staining using the indicated antibodies, and the results were quantified. **h** PLK4-myc-inducible cells were treated with siControl or siRTTN as described in **g**, and the results were quantified. The *error bars* in **f**, **g**, and **h** represent the mean ± s.d. from three independent experiments (n = 100/experiment). **i** Schematic of the interaction between RTTN and STIL. *Scale bar*, 5 μm in **f**; *Scale bar*, 0.5 μm in **g** and **h**

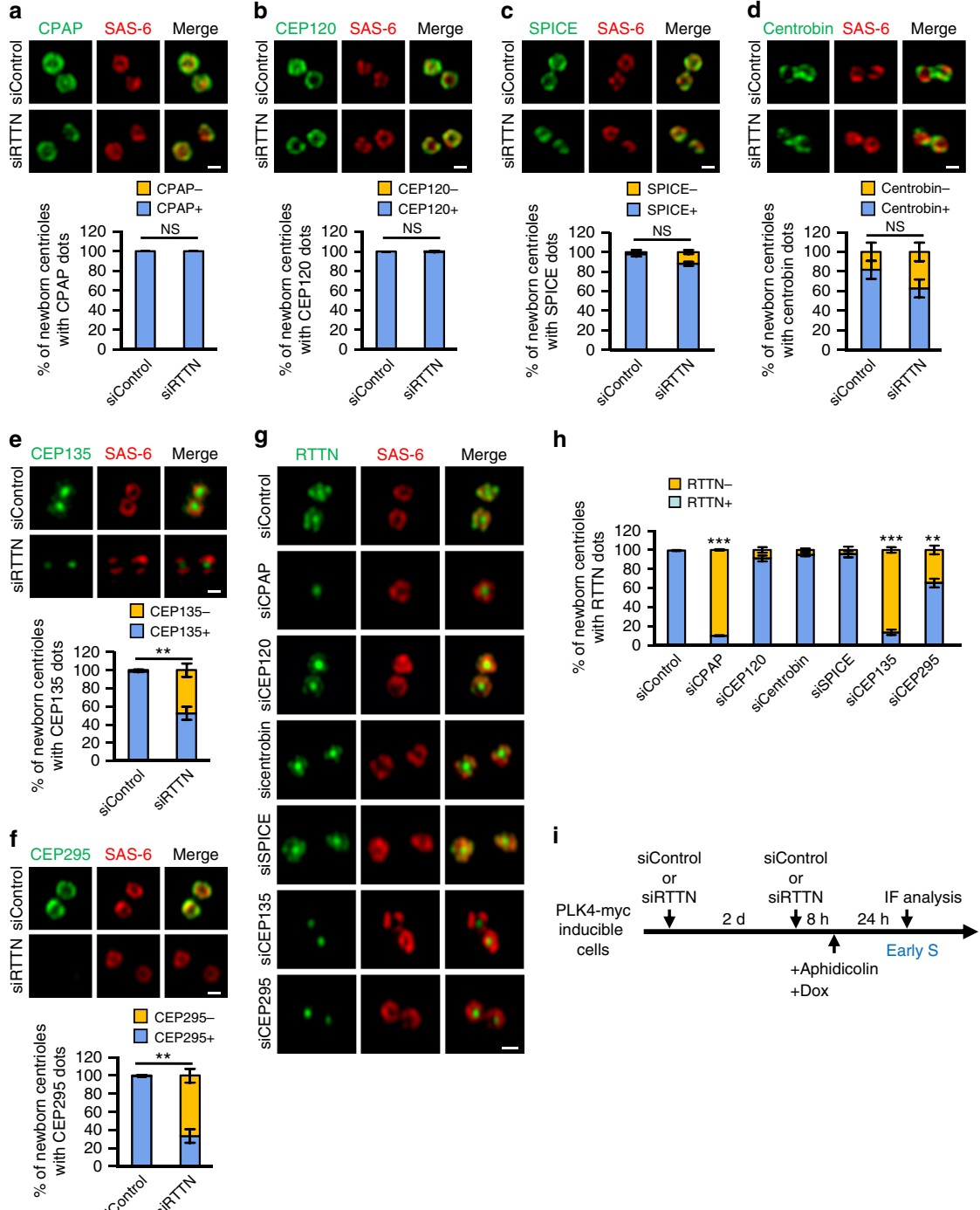

**Fig. 6** Delineation of the RTTN-mediated centriole elongation pathway. **a–f** PLK4-myc-inducible cells were treated with siControl or siRTTN as shown in **i** and analyzed by confocal fluorescence microscopy using antibodies against SAS-6 **a–f**, CPAP **a**, CEP120 **b**, SPICE **c**, centrobin **d**, CEP135 **e**, or CEP295 **f**, and the results were quantified. *Error bars* represent the mean ± s.d. (*n* = 3 independent experiments with 100 cells scored per experiment). **g** PLK4-myc-inducible cells were treated with siRNAs against CPAP, CEP120, SPICE, centrobin, CEP135, or CEP295 as shown in **i**, and analyzed by confocal fluorescence microscopy using the indicated antibodies. **h** *Histogram* illustrating the percentages of newborn centrioles with RTTN dots. *Error bars* represent the mean ± s.d. (*n* = 3 independent experiments with 100 cells scored per experiment). **P < 0.01; ***P < 0.001; NS, not significant (two-tailed *t*-test). **i** *Schematic* of the procedure used to analyze the recruitment of centriolar proteins in siRNA-treated PLK4-myc-inducible cells. *Scale bar*, 0.5 μm

**Delineation of a RTTN-mediated centriole elongation pathway.** A number of centriolar proteins, including CPAP, CEP120, SPICE, centrobin, CEP135, and CEP295, have recently been reported to be essential for centriole elongation[11–16, 19, 20]. To examine the hierarchical order of these proteins with respect to RTTN during centriole elongation, we performed siRNA-mediated depletion of these proteins in PLK4-myc-inducible cells and examined the localization of RTTN, and vice versa. Our results showed that depletion of RTTN had no effect on the localizations of CPAP, CEP120, SPICE, or centrobin to the newborn centrioles (Fig. 6a–d), but it did perturb the recruitment of CEP135 (Fig. 6e) and CEP295 (Fig. 6f) to the newborn

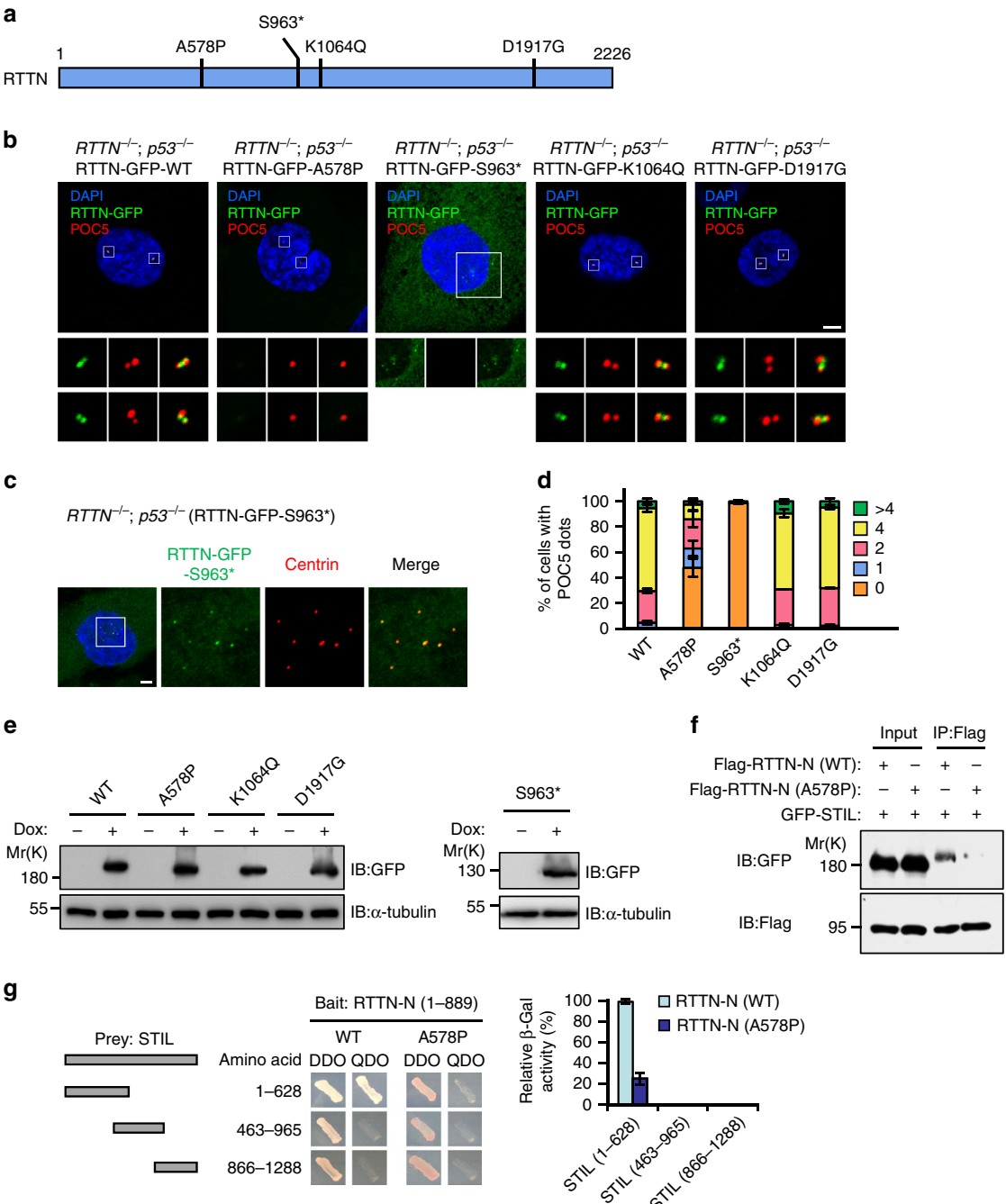

**Fig. 7** The human microcephaly RTTN mutant, A578P, exhibits reduced binding to STIL and its expression perturbs centriole duplication. **a** Schematic representation of RTTN showing the positions of RTTN mutations that cause primary microcephaly. **b, c** $RTTN^{-/-}$; $p53^{-/-}$ RPE1 cells that expressed doxycycline-inducible RTTN-GFP wild-type or mutant proteins (A578P, S963*, K1064Q, or D1917G) were synchronized by aphidicolin treatment for 24 h, and then released in fresh medium for another 15 h to allow progression to G2 phase. The cells were fixed and stained with the indicated antibodies. **d** Histogram illustrating the percentages of cells exhibiting POC5 dots. Error bars represent the mean ± s.d. from three independent experiments ($n = 100$/experiment). **e** $RTTN^{-/-}$; $p53^{-/-}$ RPE1 cells that expressed doxycycline-inducible RTTN-GFP wild-type or mutant proteins were analyzed by IB using the indicated antibodies. **f** The naturally occurring A578P mutation of RTTN reduces the binding of the protein to STIL. HEK293T cells were co-transfected Flag-RTTN-N (WT) or Flag-RTTN-N (A578P) mutant with full-length GFP-STIL. Twenty-four hours after transfection, cell lysates were IP with anti-Flag and analyzed by IB using the indicated antibodies. **g** Yeast two-hybrid assay testing interactions between various portions of STIL (prey) and truncated RTTN-N (bait, both WT and A578P). Histogram illustrating relative β-galactosidase (β-Gal) activity obtained in a liquid assay. The interaction between STIL (1–628) and RTTN-N (WT) was arbitrarily set to 100%. Error bars represent the mean ± s.d. from three independent experiments. Scale bar, 5 μm

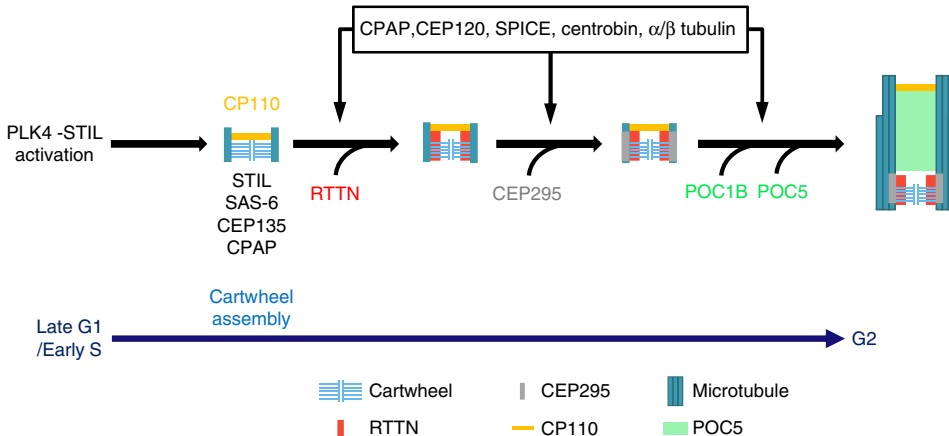

**Fig. 8** Model showing the role of RTTN in procentriole elongation. During early S phase, RTTN is recruited to the inner luminal wall of newborn centrioles, where it likely stabilizes and maintains the primitive procentrioles. RTTN then helps recruit CEP295, which mediates procentriole elongation and promotes the loading of POC5/POC1B to the distal-half centrioles. CPAP and CEP120 promote the assembly of outer 9-triplet microtubules. For details, see the Discussion. For simplicity, only one centriole is shown

centrioles. In agreement with these findings, the PPBs in RTTN-KO cells ($RTTN^{-/-}$; $p53^{-/-}$) exhibited no detectable CEP295 (Supplementary Fig. 8a), but stained positive for CEP120, SPICE, and centrobin (Supplementary Fig. 8c–e). Furthermore, the recruitment of CEP135 to the PPBs was also partially affected in RTTN-KO cells ($RTTN^{-/-}$; $p53^{-/-}$) (Supplementary Fig. 8b). Notably, our reciprocal experiments showed that depletion of CPAP and CEP135, but not CEP120, SPICE, or centrobin, substantially blocked the targeting of RTTN to the newborn centrioles, while CEP295 depletion only partially perturbed the recruitment of RTTN to the centrioles (Fig. 6h). Taken together, our data strongly support a model in which: (1) CPAP serves as an upstream effector for RTTN; (2) CEP135 and RTTN are mutual affected each localization; and (3) CEP295 acts downstream of RTTN.

**Functional characterization of RTTN mutants**. Recently, four different homozygous *RTTN* mutations (Fig. 7a), including three missense mutations (A578P, K1064Q, and D1917G) and one nonsense mutation (S963\*), were identified in patients with autosomal recessive primary microcephaly[24]. However, the molecular and cellular defects caused by these mutations remained unknown. To assess how these disease-related mutations affect the function of RTTN, we generated RPE1-based RTTN-GFP-inducible cell lines in the $RTTN^{-/-}$; $p53^{-/-}$ background, and used doxycycline to induce the expression of GFP-tagged full-length wild-type RTTN protein or various RTTN mutants.

We first examined the expression level of these proteins by western blotting with an anti-GFP antibody. No significant difference was observed in their protein expression level (Fig. 7e), indicating that the mutations did not affect protein stability. To further analyze the influence of these four mutations on RTTN localization and centriole duplication, the RTTN-GFP-inducible cells were induced to express wild-type or various RTTN mutants, synchronized in G2 phase, and analyzed by confocal immuno-fluorescence microscopy using indicated antibodies. Unexpectedly, our results showed that two RTTN mutant proteins (K1064Q and D1917G) appear to function "normally" in cells, in that they exhibited correct centrosome targeting and the cells exhibited "normal" duplication of centrioles containing the later-born centriolar protein, POC5 (Fig. 7b, d). These two mutations therefore do not appear to have any effect on centriole elongation, and we do not yet know how they trigger MCPH. In

contrast, the A578P and S963\* mutant proteins triggered abnormal phenotypes. The S963\* mutant, which introduces a premature codon in its transcribed mRNA resulted in producing a truncated protein, retained its ability to target the PPBs (Fig. 7c) but failed to convert the multiple PPBs to the normal four POC5-positive centrioles (Fig. 7d). The proper localization of this mutant to the centrosomal PPBs could reflect that the S963\* mutant carries the centrosome-targeting and STIL-interaction regions (aa 1–889; Fig. 5i). Notably, the missense mutant, A578P, lost its centrosome-targeting ability and failed to restore normal centriole duplication (Fig. 7b, d). Since the A578P mutation is located within the STIL-interacting domain of RTTN, we examined whether this mutation abrogated the interaction of RTTN with STIL. We co-transfected HEK293 T cells with vectors encoding GFP-STIL and Flag-RTTN-N (wild-type or the A578P mutant), and subjected cell lysates to co-IP assays. Our results showed that the A578P mutant exhibits a reduced binding affinity for STIL (Fig. 7f). Consistent with this finding, our yeast two-hybrid assay demonstrated only a weak interaction between STIL (1–628) and RTTN-N-A578P (Fig. 7g). Taken together, our results suggest that the decreased interaction of STIL with A578P mutant RTTN could be a precipitating cause of MCPH in humans.

**Discussion**

We are just beginning to understand molecular process through which a single procentriole grows to become a full-length centriole. Here, we characterized RTTN, which is a relatively unexplored protein that can harbor mutations recently reported to cause primary microcephaly and primordial dwarfism in humans[24]. Our results showed that RTTN: (1) directly interacts with STIL (Fig. 5); (2) localizes to the proximal inner luminal wall of centrioles (Fig. 4); and (3) is required to build distal-half centrioles (Fig. 1 and Supplementary Figs. 1 and 3). Our studies therefore show for the first time that human microcephaly protein, RTTN, plays a physiological role in building full-length centrioles. Based on our current data and that from other labs, we predict a possible model for the interplay of RTTN with other centriolar proteins involved in centriole elongation (Fig. 8).

In this proposed model, centriole assembly starts during late G1/early S phase, after PLK4-STIL activation, which triggers the assembly of a SAS-6-containing cartwheel at the proximal end of pre-existing centrioles[28–33]. CEP135 directly binds to SAS-6 and CPAP, linking the cartwheel to the outer microtubules[9]. CPAP

then cooperates with additional proteins, such as CEP120 and SPICE to promote the assembly of centriolar microtubules, resulting in the growth of procentrioles during S/G2 phase[14, 15]. During early S phase, RTTN is recruited to the inner luminal wall of newborn centrioles, where it likely stabilizes and maintains the primitive procentrioles that contain STIL, CPAP, and SAS-6-containing cartwheel (Supplementary Fig. 3d–f). These RTTN-stabilized primitive procentrioles are required for the proper loading of later-born centriolar proteins (e.g., POC5 and POC1B) to the distal-half centrioles at a later stage (Supplementary Figs. 1 and 3). Consistent with this model, we found that depletion or loss of RTTN did not affect the localization of early-born centriolar proteins (SAS-6, STIL, CPAP, or CP110) to the newborn centrioles, but severely affected the recruitment of POC5 and POC1B to the distal-half centrioles. Furthermore, the unclear procentriole EM images obtained in siRTTN-treated cells (Fig. 3f, lower panel) may possibly represent procentrioles that have been destabilized by the loss of RTTN. Last, the centriolar MT-interacting protein, CEP295, was recently reported to be essential the recruitment of POC5 and POC1B to the distal portion of centrioles[19]. Our present results show that loss of RTTN completely blocked the targeting of CEP295 to PPBs (Supplementary Fig. 8a), but CEP295 depletion only partially affected RTTN localization (Fig. 6h). These findings strongly support the idea that RTTN acts as an upstream effector of CEP295, which mediates the loading of later-born centriolar proteins to the distal-portion centrioles at a later stage of centriole biogenesis.

STIL was previously reported to directly interact with CPAP[8], which works with CEP120 to promote centriole elongation[14, 15]. Here, we found that STIL could also interact with RTTN, and that this interaction is required for the CEP295-mediated loading of POC5 and POC1B to the distal-half centrioles. Thus, STIL plays critical roles in both: (1) the assembly of SAS-6-containing cartwheels after the activation of PLK4 at the proximal end of newborn centrioles; (2) procentriole elongation and the loading of later-born centriolar proteins to distal-half centrioles possibly via the STIL-CPAP[8] and STIL-RTTN (this report) interactions. Finally, the localizations of CEP135 and RTTN seem to be mutually affected by one another, since depletion of CEP135 severely blocks the recruitment of RTTN to the centrioles (Fig. 6h), while complete loss of RTTN also interferes (partially) with the targeting of CEP135 to PPBs (Supplementary Fig. 8b).

Autosomal recessive primary microcephaly (MCPH) is a neurodevelopmental disorder characterized by marked brain size reduction and mental retardation[34–36]. A number of genes that have been implicated in MCPH (e.g., *PLK4, SAS-6, STIL, CPAP/CENPJ, CDK5RAP2, CEP135, CEP152*, and *CEP63*) encode proteins that localize to centrioles or centrosomes[37–43]. Interestingly, direct interactions among these microcephaly proteins, including the CEP63-CEP152[39], CEP152-CPAP[44, 45], STIL-CPAP[8], and CPAP-CEP135-hSAS-6[9] interactions, reportedly participate in the early onset of cartwheel assembly and procentriole formation. Here, we show that the RTTN-STIL interaction is not essential for the early onset of procentriole formation, but rather is later required to build full-length centrioles via the loading of later-born centriolar proteins to the distal-half centrioles. Collectively, the present and prior findings suggest that maintenance of intact functional centrioles is critical for the survival of neural progenitor cells, since mutations in genes whose products participate in the assembly of functional centrioles cause MCPH in humans, and the absence of functional centrioles could induce a p53-dependent neuronal cell death[46]. Currently, it is not yet known why MCPH patients show more severe clinical symptoms in the brain compared to other tissues. It is possible that some unique features of neural stem cells

(such as neuronal cell division, migration, apical-basal polarity, Interkinetic nuclear migration, and/or cilia formation) depend heavily on the normal function of intact centrioles. However, the main roles of centrioles in the above features have not yet been fully explored in neural progenitor cells. Future comparative studies of such features in normal and acentriolar neural stem cells (in $p53^{-/-}$ background) of both wild-type and conditional knockout mice or in human iPS cell-derived brain organoids of normal and microcephaly patients may help reveal the underlying mechanisms.

Finally, it is interesting to note that the complete loss of RTTN in p53-null cells induced the de novo formation of multiple PPBs that contained many early-born procentriolar proteins (e.g., SAS-6, STIL, CPAP, and so on), but lacked later-born proteins (POC5, POC1B) (Supplementary Fig. 3). However, exogenous expression of wild-type RTTN rescued this phenotype and converted the multiple PPB phenotype to a normal centriole number (2 or 4 centrioles per cell) (Supplementary Fig. 3b-iii and Fig. 7b). A similar phenotype was previously observed in CEP295-knockout cells[47]. Here, we show that RTTN appears to be an upstream effector of CEP295, and both proteins participate in later steps of centriole elongation (e.g., recruitment of POC5/POC1B to the distal portion of centrioles). Thus, in addition to centriole elongation, RTTN and CEP295 may also play some yet-uncharacterized role(s) in the formation of PPBs in the absence of p53. It seems that RTTN and CEP295 might act as guides to monitor the procentriole copy number. Notably, CEP295 was recently reported to play an essential role in converting the daughter-to-mother centriole[47–49]. Thus, experiments aimed at unraveling the potential roles of RTTN and CEP295 in regulating de novo PPBs formation and centriole-to-centrosome conversion will be exciting topics for future study.

## Methods

**Plasmids.** The human RTTN cDNA (accession number, BC156291) was purchased from the IMAGE clone consortium (IMAGE clone No. 100061722) and subcloned into pEGFP-N2 (BD Biosciences Clontech) or pFlag-CMV2 (Sigma-Aldrich). To generate the GST-fusion constructs, the cDNAs encoding various potion of RTTN were fused in-frame to the GST-encoding sequence in the pGEX4T vector (GE Healthcare). The RTTN mutant constructs were generated by site-directed mutagenesis using a QuikChange kit (Stratagene). The expression constructs encoding various GFP-tagged RTTN mutant proteins were generated using pcDNA4/To/myc-His-A (Invitrogen) or pLVX-Tight-Puro (BD Biosciences Clontech) vectors. Sequencing was used to confirm all generated plasmids. The cDNAs encoding full-length CP110, H2B and STIL were obtained by RT-PCR using total RNAs from human HEK293T cells and subcloned in-flame into a pEGFP-C1 (BD Biosciences Clontech) or a pLVX-Tight-Puro (BD Biosciences Clontech) vector. The cDNA constructs for GFP-tagged full-length STIL, SAS-6, and CPAP and various Flag-tagged STIL truncated mutants were from ref. [8]. The cDNA constructs containing various CEP295 fragments were from ref. [19].

**Antibodies.** The rabbit polyclonal antibody against RTTN was raised using recombinant RTTN-His (residues 1347–1591) and subjected to affinity purification. The following antibodies were used in this studies: antibodies against SAS-6 (residues 1–489, 1:500 dilution)[13], CPAP (residues 1070–1338, 1:1000 dilution)[50], CEP120 (residues 639–986, 1:1000 dilution)[15], CEP135 (residues 650–1140, 1:1000 dilution)[9], CEP295 (residues 2092–2430, 1:500 dilution)[19], centrobin (residues 443–626, 1:1000 dilution)[13], and centrin 2 (residues 1–173, 1:1000 dilution)[13]. The other commercially available antibodies used in this study included anti-STIL (A302–442; 1:200 dilution), anti-hPOC5 (A303–341; 1:500 dilution) and anti-SPICE (A303–272; 1:500 dilution) (all from Bethyl); anti-hSAS-6 (H00163786; 1:100 dilution, monoclonal Ab), anti-centrin3 (H00001070-M01; 1:1000 dilution) and anti-CEP162 (PAB22408; 1:1000 dilution) (all from Abnova); anti-CP110 (12780-1-1p; 1:500 dilution; Proteintech); anti-CEP164 (NBP1-81445; 1:500 dilution; Novus Biologicals); anti-POC1B (PA5-24495; 1:500 dilution; Thermo Fisher); anti-acetylated tubulin (T6793; 1:1000 dilution) and anti-Flag (M2; F3165; 1:3000 dilution) (both from Sigma-Aldrich); and anti-GFP (632381; 1:3000 dilution; BD Bioscience).

**Cell culture and transfection.** U2OS and HEK293T cells, originally obtained from ATCC, were grown in Dulbecco's modified Eagle's (DME) medium supplemented

with 10% fetal calf serum. Human telomerase-immortalized retinal pigment epithelial cells (hTERT-RPE1 and $p53^{-/-}$ hTERT-RPE1)[51] were maintained in DME/F12 (1:1) medium supplemented with 10% FBS. Cells were transfected with various cDNA constructs using Lipofectamine 2000 (Invitrogen). For synchronization experiments, cells were arrested at early S phase by incubation with 2 µg ml$^{-1}$ aphidicolin for 24 h, and then released in fresh medium for another 15 h to enrich for G2-phase cells[8]. For PLK4 inhibition experiments, cells were treated with 300 nM centrinone B[26] (5690; TOCRIS Bioscience). All cell lines were tested without mycoplasma contamination.

**Production of RTTN$^{-/-}$; p53$^{-/-}$ cells by the CRISPR system**. Because RTTN$^{-/-}$ RPE1 cells cannot survive in the presence of wild-type p53, we generated RTTN$^{-/-}$; $p53^{-/-}$ RPE1 cells in which both RTTN and p53 genes were mutated using the CRISPR-mediated gene targeting system[52]. RNA-guided targeting of RTTN in hTERT-RPE1 cells was achieved through coexpression of the Cas9 protein with gRNAs. The gRNA expression plasmids were generated by inserting annealed primers into the gRNA cloning vector (plasmid #41824; Addgene). The targeting sequences for the RTTN gRNAs were 5′-CACCAACGTCAACCAAATGT-3′ and 5′-TCCCCCAGCAGTCCAACATT-3′. For nucleofection, 2.5 µg hCas9 plasmid (plasmid #41815; Addgene) and 2.5 µg gRNA were mixed with electroporation buffer (Lonza) according to the manufacturer's instructions, and then used to treat $p53^{-/-}$ hTERT-RPE1 cells[47, 51]. The nucleofected cells were serially diluted, and single colonies were picked and expanded. The protein expression levels of RTTN in the picked colonies were examined by immunofluorescence microscopy or Western blotting to confirm the loss of RTTN. Genomic DNAs isolated from RTTN-knockout colonies were subjected to PCR using the following primers: 5′-GGCTCATCTTTCATAAATGTTATCAAG-3′ and 5′-GAGGAGGGGAAAA AGGTCAAAATC-3′ for clone #1 and clone #2. The PCR products were cloned and sequenced.

**Generation of doxycycline-inducible cell lines**. The U2OS-based doxycycline-inducible PLK4-myc[19], CPAP-myc[13], and CEP120-myc[15] cell lines were used in this study. To generate the U2OS-based RTTN-GFP-inducible line, the cDNA fragment encoding RTTN-GFP was subcloned into the pcDNA4/To/myc-His-A vector (Invitrogen), which placed it under the control of doxycycline. The construct was transfected into U2OS-T-Rex cells stably expressing the Tet repressor TetR, using Lipofectamine 2000 (Invitrogen). The expressions of the PLK4-myc, CPAP-myc, CEP120-myc, or RTTN-GFP were induced by adding 1 µg ml$^{-1}$ doxycycline to the culture medium. To obtain RTTN$^{-/-}$; p53$^{-/-}$ RPE1 cell lines inducibly expressing RTTN-GFP (wild-type or mutants), GFP-CP110, mCherry-H2B or mCherry-STIL, lentiviruses containing RTTN-GFP (wild-type or mutants), GFP-CP110, mCherry-H2B, or mCherry-STIL in the pLVX-tight-puro vector[51] were used to infect RTTN$^{-/-}$; $p53^{-/-}$ RPE1 Tet-On cells (stably expressing rtTA), and the cells were sterile-sorted by flow cytometry for RTTN-GFP (wild-type or mutant), GFP-CP110, mCherry-H2B, or mCherry-STIL.

**siRNA analysis**. All siRNAs and the non-targeting siRNA control were obtained from Invitrogen (siRNA sequences detailed in Supplementary Table 1), and transfections were performed using Lipofectamine RNAiMAX (Invitrogen) according to the manufacturer's protocol. All RTTN siRNAs were found to work well; we used siRTTN#1 for the reported experiments.

**Immunoprecipitation and GST pulldown**. HEK293T cells were transiently transfected with various constructs. Twenty-four hours after transfection, cells were lysed in NP-40 lysis buffer (50 mM Tris-HCl at pH 8.0, 150 mM NaCl, 1% NP-40, 20 mM β-glycerophosphate, 20 mM NaF, 1 mM Na₃VO₄, with protease inhibitors, including 1 µg µl$^{-1}$ leupeptin, 1 µg µl$^{-1}$ pepstatin, and 1 µg µl$^{-1}$ aprotinin). The cell lysates were immunoprecipitated with the indicated antibodies for 2 h at 4 °C and then incubated with protein-G-sepharose beads for another 2 h. The immunoprecipitated complexes were separated by SDS-PAGE and probed with the indicated antibodies.

For GST pulldown experiments, HEK293T cells were transiently transfected with various constructs. Twenty-four hours after transfection, the cells were lysed in NP-40 lysis buffer as described above. The GST-RTTN-N recombinant protein was expressed in *Escherichia coli* by IPTG induction, and affinity purification was performed with glutathione-agarose beads (Sigma-Aldrich). Cell lysates were incubated with immobilized GST-RTTN-N at 4 °C for 3 h. After incubation, the samples were washed with NP-40 lysis buffer, separated by SDS-PAGE, and examined by western blot analysis. All uncropped western blots can be found in Supplementary Figs. 9–11.

**Yeast two-hybrid assay**. Yeast two-hybrid analysis, which was used to demonstrate a direct protein–protein interaction, was performed using the Matchmaker Gold yeast two-hybrid system (Clontech)[8]. Briefly, the cDNA fragments of STIL and RTTN were subcloned into either pGBKT7 as baits or pGADT7 as preys. The constructs were subsequently transformed into Y2HGold and Y187 yeast stains. After mating, the diploid yeasts were grown on both DDO (SD minimal medium, −Trp, −Leu) and QDO (SD minimal medium, −Trp, −Leu, −Ade, −His) plates at 30 °C. After 3–4 days incubation, positive colonies were

picked and restreaked to obtained isolated colonies. Two independent colonies were streaked per sample. The activation of β-galactosidase (β-Gal) by protein interactions was assayed using o-nitrophenol-β-galactopyranoside (ONPG) as a substrate with yeast diploids grown in DDO medium.

**Microscopy**. For immunofluorescence confocal microscopy and super-resolution microscopy (3D-SIM), cells were grown on coverslips, cold treated for 1 h at 4 °C, and then fixed in methanol at −20 °C for 10 min[19]. The cells were blocked with 10% normal goat serum in PBST, incubated with the indicated primary antibodies, washed, and then incubated with Alexa Fluor 488-, Alexa Fluor 568-, or Alexa Fluor 647-conjugated secondary antibodies (Invitrogen). Cell proliferation analysis and EdU labeling were carried out using the Click-iT EdU Imaging Kit with Alexa Fluor (Life Technologies) according to the manufacturer's instructions. DNA was counterstained with DAPI. The samples were mounted in Vectashield mounting media (Vector Laboratories), and visualized using a LSM 780 system or a LSM880 Airyscan system (Carl Zeiss) with a Plan Apochromat ×100 (1.4 NA) oil-immersion objective. Super-resolution images were obtained using a Zeiss ELYRA system (Carl Zeiss) with a Plan Apochromatic ×63 (1.4 NA) oil-immersion objective. The percentages of newborn centrioles with various centriolar proteins (Fig. 6 and Supplementary Fig. 1) were quantified by the presence or absence of the indicated protein signals on the ring-like structures of siControl- and siRNA-treated cells.

For Immunogold study, U2OS cells were fixed and examined by electron microscopy (EM)[19]. Briefly, cells were grown on Aclar film (Electron Microscopy Sciences), fixed with 3% paraformaldehyde/2% sucrose for 10 min, permeabilized with 0.5% TritonX-100 for 2 min, and blocked with 1% BSA for 30 min. The cells were incubated with anti-RTTN overnight at 4 °C, washed, and then incubated with goat anti-rabbit IgG-Nanogold (1:40; Nanoprobes) for 1 h at room temperature (RT). The labeled cells were further fixed with 2.5% glutaraldehyde in PBS for 1 h, silver-enhanced for 4 min using the HQ silver reagent (Nanoprobes), washed with distilled water, dehydrated in ethanol, and embedded in epoxy resin.

For EM, cells grown on Aclar film (Electron Microscopy Sciences) were fixed in 2.5% glutaraldehyde with 1% tannic acid in 0.1 M cacodylate buffer[19]. The cells were post-fixed in 1% OsO₄ in 0.1 M cacodylate buffer at RT for 30 min, stained with 1% uranyl acetate at RT for 1 h, dehydrated in a graded series of ethanol, infiltrated, and embedded in Spurr's resin. Thin sections (80 nm) were stained with 4% uranyl acetate and Reynold's lead citrate for 10 min, and examined with an electron microscope (T FEG-TEM; FEI Tecnai G2 TF20 Super TWIN).

**Live-cell imaging**. Time-lapse live-cell movies were taken with a LSM 780 confocal microscope (Carl Zeiss) using a Plan Apochromat ×100 (1.4 NA) oil-immersion objective in a 37 °C incubator chamber with 5% CO₂. The Z-stacks were reconstructed with the Zen software (Carl Zeiss).

**Statistical analysis**. Results are presented as mean plus s.d. as specified in each figure legend. Statistical differences between two data sets were analyzed using the two-tailed unpaired Student's *t*-test (GraphPad Prism 5); *$P < 0.05$, **$P < 0.001$, and ***$P < 0.0001$ were considered statistically significant.

**Data availability**. All data supporting the findings of this study can be found within the paper and its Supplementary Information files, or are available from the corresponding author upon request.

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

## Acknowledgements

We thank the sequencing core facility (IBMS), the confocal imaging core facilities (IBMS, SIC, ABRC, NPAS), and the EM core facilities (IMB, SIC) of Academia Sinica. We thank Ching-Wen Chang for helpful discussion and Shu-Chen Shen for technical assistance. This work was supported by grants from the Ministry of Science and Technology, Taiwan, ROC (MOST 105-2321-B001-016) and the Academia Sinica Investigator Award.

## Author contributions

H.-Y.C., a PhD student at the Graduate Institute of Life Sciences at the National Defense Medical Center, performed most of the experiments, designed the study, interpreted data, and wrote the initial draft of the manuscript. C.-T.W., C.-J.C.T., Y.-N.L., and W.-J.W. performed experiments. T.K.T. conceived and designed the study, interpreted the data, and wrote the manuscript.

## Additional information

**Competing interests:** The authors declare no competing financial interests.

