## [Peer Review File · Nature Communications]

Reviewers' comments:

Reviewer #1 (Remarks to the Author):

Chen and colleagues address the molecular roles of a novel microcephaly protein, Rotatin (RTTN), in centriole assembly. The molecular functions of human RTTN are so far poorly understood, while its *Drosophila* homologue, Ana3, is known to be important for the stability of centrioles, but is not required for their duplication. In this study the authors show that human RTTN is dispensable for the initial stages of centriole assembly, but is important for the subsequent elongation and maturation of new centrioles. They address the spatial and temporal localization pattern of RTTN at centrosomes and its molecular and functional interactions with several key centriolar components. Most interestingly, the authors find that RTTN interacts directly with the key centriolar component STIL, and that a naturally occurring mutation identified in patients with microcephaly weakens this interaction and results in mislocalization of RTTN from centrioles.

The results presented in this study are important for the field. Although previously linked to genetic disease, the molecular roles of RTTN have so far been under-characterized and therefore these data present a significant advance. The experiments are well presented and the conclusions are mostly supported by data from multiple experimental approaches, which greatly strengthens the paper. Below I list a few suggestions for the authors and some points that I think require clarification. Once these have been addressed, I would strongly support publication of this work in *Nature Communications*.

Major Points:

1. The authors imply that they needed to make the RTTN knockouts in a p53^{-/-} background because acentriolar cells are not viable in a p53⁺ background. Did they try making these cells in a p53⁺ background first and failed, or did they only try in p53^{-/-}? This seems an important point, as these cells clearly can make structures that have some similarity to centrioles (see point 2, below). Do these structures have any residual function, or do RTTN^{-/-} cells behave in the same way as cells completely lacking centrioles (e.g. when Plk4 is inactivated)?
2. The authors describe that RTTN^{-/-};p53^{-/-} knockout cells do not contain typical centrioles but instead produce many de novo, short, procentriole-like structures in early S-phase that then later disassemble in mitosis. The description of this process is a bit vague. Do these structures completely disappear, so that cells proceed through mitosis with no centrioles or centrosomes at all? If so, do the mitotic cells exhibit defects typical of cells that lack centrosomes? The authors should comment on why they think these de novo centrioles might disappear during mitosis.
3. It would be interesting to know whether normal centrioles are gradually destabilised in RTTN deficient cells. This could be addressed by following the fate of daughter centrioles in siRTTN cells (either normal or PLK4-induced).
4. In contrast to the clear, petal-like arrangement of individual daughter centrioles in siCtrl PLK4-inducible cells, in siRTTN PLK4-inducible cells the centriole markers labelling daughter centrioles consistently appear as continuous rings around the mother centriole (Suppl. Fig. 4, Figure 2, Figure 5). The authors should comment on this unexplained phenomenon.
5. Related to the point above, it is unclear how percentage values of new born centrioles can be quantified in the Figures listed above, as the individual daughter centrioles do not appear to be resolvable from each other in the images shown. The authors should supply a short explanation of this in the Methods section.
6. The data presented in Figure 4f showing the centrosomal localization of the 3 different GFP-RTTN fragments analysed should be quantified.

7. The authors conclude that depletion of STIL results in strongly diminished localization of RTTN to newborn centrioles in PLK4-inducible cells (Figure 4g). I would have thought that in this experiment, where STIL is depleted prior to S-phase synchronization, newborn centrioles are unlikely to form at all, and there is no indication of their presence in the data shown. Unless the authors can show that centrioles are present but they cannot recruit RTTN, I would suggest toning down the conclusions of this experiment.

8. The authors should include quantitative data to support the localization pattern of RTTN at centrioles during the cell cycle, to support Figure 3a and the conclusions described in the main text (page 6).

Minor Points:

1. On page 9, it is unclear what the asterisk in 'RTTN-GFP*-inducible cell lines' stands for, as this sentence refers collectively to all 4 patient-mutations the authors analysed.

2. In Supplementary Figure 2 I would recommend labelling the cell cycle stages above the images, or specifying what stage t=0mins relates to within the figure legend.

3. The authors introduce the acronym 'PCL' for the 'procentriole-like structures' that they observe in RTTN knockout cells. This acronym is already used in the centriole field for proximal centriole-like structures in insect spermatids (Blachon et al. Genetics 2009). I would recommend choosing a different definition to avoid future confusion of these terms.

Reviewer #2 (Remarks to the Author):

The manuscript by Chen et al is an exceptionally well designed and high quality study, with clear conclusions about the functional hierarchy of proteins in assembling centrioles, particularly related to the building of full-length centrioles. The authors have used multiple high-end approaches (such as CRISPR knock out cells lines, beautiful confocal and super resolution microscopy imaging, and immunoelectron microscopy) to substantiate their conclusions on centriolar assembly. The paper is well written, and is an easy read. The model at the end of the paper, and the small cartoons in some other figures help in interpreting the data effectively. The functional characterization of human microcephaly mutations in inducible cell culture model systems would require further study in more physiologically relevant contexts using neural stem/progenitor cells, and in vivo approaches such as conditional knock out/in models (both beyond the scope of this paper). The present study is an important advance in our current understanding of how centrioles are assembled, and will be an important addition to our existing knowledge on centriole assembly after formation of cartwheels. I am happy to accept the paper as it is, without any further modifications and delay in publishing.

I have a few minor suggestions:

_The authors might consider adding statistics to some of the figures, although the data is very easily interpretable in most cases. The only figure that begs a bit of explanation is 5d, where both SPICE and Centrobin recruitment to centrioles seem to be partially affected upon RTTN knockdown.

_The comment in discussion on dwarfism resulting from malfunctioning in cell divisions throughout

the body might be carefully considered.

Reviewer #3 (Remarks to the Author):

This manuscript by Chen et al explores the role of the protein Rotatin (RTTN) in centriole biogenesis. RTTN is the human ortholog of the *Drosophila* protein Ana3, a centriole protein identified in 2007 by the Vale lab. Since then, only one study has explored Ana3's role at centrioles, and no lab has yet studied RTTN. For this reason, along with its link to human microcephaly, make the topic very interesting. However, as you will see from my comments below, I do not support the publication of this work in Nature Communications. The observations are preliminary and the presented model is unsupported.

Major comments

1. I do not agree with the conclusion that RTTN is required for centriole elongation. I do not see any evidence that a procentriole is present in the knockdown, which then does not elongate. All of the presented data suggest that RTTN plays a role early in centriole duplication. In fact, the authors state that RTTN is involved in duplication later in the manuscript. All the images presented, both light microscopy and EM, show disorganized protein masses around the mother centriole in RTTN RNAi. It seems to me that RTTN is required for properly organizing proteins early in duplication. For example, figure 5a-d show abnormal localization of all of these proteins. The most striking is the localization of Sas6 in Figure 5 and S4b, where the WT Sas6 are clear dots, the RTTN RNAi Sas6 is a smear. Yet, the authors use a + / - readout to determine proper organization. In addition, the localization of RTTN is proximal, and it is not clear how a proximal protein plays a role in elongation per se.

Related to the above, the overexpressed CPAP-myc experiment can also be explained by improper organization of proteins during early duplication. If early events are messed up, then naturally elongation will not occur either. Their conclusion does not match the data.

2. The use of the Procentriole-Like must be removed. The presence of cytoplasmic protein dots should not be compared to a procentriole in any way unless there is evidence of some semblance of procentriole structure. It is not surprising that centriole proteins appear in the cytoplasm when one has removed its normal docking site. It is quite interesting that the aggregation in the cytoplasm is still controlled throughout the cell cycle, but this does not prove that RTTN controls centriole elongation.

In the end, it is clear that RTTN is important for centriole biogenesis, but it is not clear what RTTN does. The authors offer little insights into the mechanism of RTTN function, and their general conclusion of elongation is not supported. It is my opinion that this work is not suitable for Nature Communications.

Reviewer #1:

Chen and colleagues address the molecular roles of a novel microcephaly protein, Rotatin (RTTN), in centriole assembly. The molecular functions of human RTTN are so far poorly understood, while its Drosophila homologue, Ana3, is known to be important for the stability of centrioles, but is not required for their duplication. In this study the authors show that human RTTN is dispensable for the initial stages of centriole assembly, but is important for the subsequent elongation and maturation of new centrioles. They address the spatial and temporal localization pattern of RTTN at centrosomes and its molecular and functional interactions with several key centriolar components. Most interestingly, the authors find that RTTN interacts directly with the key centriolar component STIL, and that a naturally occurring mutation identified in patients with microcephaly weakens this interaction and results in mislocalization of RTTN from centrioles.

The results presented in this study are important for the field. Although previously linked to genetic disease, the molecular roles of RTTN have so far been under-characterized and therefore these data present a significant advance. The experiments are well presented and the conclusions are mostly supported by data from multiple experimental approaches, which greatly strengthens the paper. Below I list a few suggestions for the authors and some points that I think require clarification. Once these have been addressed, I would strongly support publication of this work in Nature Communications.

Major points:

1). The authors imply that they needed to make the RTTN knockouts in a p53^{-/-} background because acentriolar cells are not viable in a p53⁺ background. Did they try making these cells in a p53⁺ background first and failed, or did they only try in p53^{-/-}? This seems an important point, as these cells clearly can make structures that have some similarity to centrioles (see point 2, below). Do these structures have any residual function, or do RTTN^{-/-} cells behave in the same way as cells completely lacking centrioles (e.g. when Plk4 is inactivated)?

Response: We did first try to make the RTTN knockouts in a p53^{+/+} background, but these efforts unfortunately failed. We thus switched to a p53^{-/-} background, in which we successfully obtained two independent RTTN-knockout lines. Since RPE1 is a near-diploid human cell line with a modal chromosome number of 46, we detected no primitive procentriole bodies (PPBs) containing STIL, SAS-6, or CPAP in RPE1-based RTTN^{+/+}; p53^{-/-} cells (Fig. 1d-f). The PPB is an update of the term "PCL" used in the previous version, see below. In the revised manuscript, we present new data showing that the depletion of PLK4 or addition of centrinone B (a PLK4 inhibitor) could inhibit the formation of PPBs in RTTN^{-/-}; p53^{-/-} cells (Supplementary Fig. 3).

2). The authors describe that *RTTN*^{-/-};*p53*^{-/-} knockout cells do not contain typical centrioles but instead produce many *de novo*, short, procentriole-like structures in early S-phase that then later disassemble in mitosis. The description of this process is a bit vague. Do these structures completely disappear, so that cells proceed through mitosis with no centrioles or centrosomes at all? If so, do the mitotic cells exhibit defects typical of cells that lack centrosomes? The authors should comment on why they think these *de novo* centrioles might disappear during mitosis.

Response: Our live-cell imaging studies showed that, these PPBs largely (if not completely) disappeared in *RTTN*^{-/-}; *p53*^{-/-} cells during mitosis (Supplementary Fig. 2). Our group and others previously showed that both STIL and SAS-6 are cell cycle-regulated proteins (refs 6 and 8), and few to no STIL/SAS-6 signals were detected in mitotic cells (including metaphase and anaphase). Thus, it is possible that PPBs were not formed in the absence of STIL and SAS-6 during mitosis. This would argue that the *RTTN*-knockout cells might exhibit abnormal mitotic phenotypes that lack centrioles/centrosomes. Indeed, we observed abnormal bipolar, monopolar, and multipolar spindles in *RTTN*-knockout cells (see Figure below). We previously reported a very similar pattern in CPAP-depleted cells, which contained few to no centrioles, exhibited a prolonged mitotic delay, and had multiple mitotic abnormalities, including abnormal bipolar, monopolar, and multipolar spindles (Chou et al., Cell Report, 2016). In the revised manuscript, we have added an explanation (p4) as to why the *de novo* centrioles might disappear during mitosis.

Figure. In the *RTTN*^{-/-}; *p53*^{-/-} cells that lack centrioles, the complete loss of *RTTN* is associated with abnormal mitotic phenotypes.

3). It would be interesting to know whether normal centrioles are gradually destabilised in *RTTN* deficient cells. This could be addressed by following the fate of daughter centrioles in *siRTTN* cells (either normal or *PLK4*-induced).

Response: We used time-lapse live-cell imaging to trace the fate of daughter centrioles in siRTTN-treated RPE1 cells expressing GFP-centrin. However, the background signal noise and the resolution limit of live-cell microscopy made it difficult to follow the fate of daughter centrioles in siRTTN-treated cells.

4). *In contrast to the clear, petal-like arrangement of individual daughter centrioles in siCtrl PLK4-inducible cells, in siRTTN PLK4-inducible cells the centriole markers labelling daughter centrioles consistently appear as continuous rings around the mother centriole (Suppl. Fig. 4, Figure 2, Figure 5). The authors should comment on this unexplained phenomenon.*

Response: We have added new evidence (Supplementary Fig. 6) to explain this phenomenon. As the procentriole is known to elongate during S-G2 phases, we carefully examined the elongation of procentrioles labeled with antibodies against SAS-6 and CP110 in PLK4-myc-inducible cells during S-G2 phases. The PLK4-myc-inducible cells were treated with siControl or siRTTN, exposed to aphidicolin, and then released at different time points, as described in Supplementary Fig. 6e. The cells were then immunostained with the indicated antibodies and examined by confocal fluorescence microscopy. To estimate the procentriole length, the distance between the SAS-6 (a proximal end protein) and CP110 (a distal end protein) signals was quantified in PLK4-induced centrioles (Supplementary Fig. 6b). At the early stages of centriole elongation (0 h, early S), the newly formed CP110- and SAS-6-labeled procentrioles were observed as ring-like structures. As elongation proceeds, these ring-like structures gradually elongated (as measured by the distance between the SAS-6 and CP110 signals) to become a clear rosette pattern (15h) wherein each petal may represent an elongated centriole. A similar observation was also reported by Comartin et al. (ref. 14). In contrast, depletion of RTTN significantly blocked this procentriole elongation, often producing a smear of ring-like structures in PLK4-myc inducible cells (Supplementary Fig. 6c). Together, these findings could explain why we frequently observed a smeared ring-like structure rather than distinct dots at the early stages of centriole elongation in PLK4-myc-inducible cells. Furthermore, our findings could also explain why depletion of RTTN commonly produced a smeared ring-like structure (not a distinct petal) (Supplementary Fig. 5a-c) and shorter centrioles in PLK4-inducible cells (Fig. 2). These results strongly support our contention that RTTN contributes to centriole elongation. We have replaced the inappropriate images with the new images and commented this effect in the revised manuscript (p5-6).

5). *Related to the point above, it is unclear how percentage values of new born centrioles can be quantified in the Figures listed above, as the individual daughter centrioles do not appear to be resolvable from each other in the images shown. The authors should supply a short explanation of this in the Methods section.*

Response: The percentages of new born centrioles with various centriolar proteins (Fig. 5 and Supplementary Fig. 5) were quantified by the presence or absence of the indicated protein signals on

the ring-like structures of siControl- and siRNA-treated cells. We have included a short sentence in the Methods section (p17) of the revised manuscript.

6). *The data presented in Figure 4f showing the centrosomal localization of the 3 different GFP-RTTN fragments analysed should be quantified.*

Response: As the reviewer suggested, the quantitative results showing the centrosomal localization of the three different GFP-RTTN fragments have been added to Figure 4f of the revised manuscript.

7). *The authors conclude that depletion of STIL results in strongly diminished localization of RTTN to newborn centrioles in PLK4-inducible cells (Figure 4g). I would have thought that in this experiment, where STIL is depleted prior to S-phase synchronization, newborn centrioles are unlikely to form at all, and there is no indication of their presence in the data shown. Unless the authors can show that centrioles are present but they cannot recruit RTTN, I would suggest toning down the conclusions of this experiment.*

Response: We appreciate the reviewer's suggestion, and have toned down our conclusions related to this experiment (p9 in the revised manuscript), as follows:

"Our data thus collectively suggest that RTTN is a downstream effector of STIL, and that RTTN is very likely to be recruited to the proximal end of newborn centrioles via the N-terminal domain of STIL at early S phase. However, since depletion of STIL completely blocked the formation of newborn centrioles, further experiments are needed to test the latter hypothesis."

8). *The authors should include quantitative data to support the localization pattern of RTTN at centrioles during the cell cycle, to support Figure 3a and the conclusions described in the main text (page 6).*

Response: As suggested by the reviewer, we have added quantitative data to Figure 3a. We also describe our conclusions in the revised manuscript (see p7), as follows:

"The intensities of RTTN at the mother centriole (MC) and procentriole (PC) were quantified (Fig. 3a). Our immunofluorescence analyses showed that RTTN was associated with both centrioles during G1 phase. Interestingly, the intensity of RTTN at the PC, but not the MC, gradually increased during S phase (EdU-positive cells) and finally reached a plateau at G2/M (Fig. 3a)."

Minor points:

1). On page 9, it is unclear what the asterisk in 'RTTN-GFP*-inducible cell lines' stands for, as this sentence refers collectively to all 4 patient-mutations the authors analysed.

Response: The term "RTTN-GFP*-inducible cell lines" refer to cells harboring the four RTTN mutations found in patients. To avoid confusion with the S963* mutant, we reworded the sentence (p9, in the revised manuscript) as follows:

"To assess how these disease-related mutations affect the function of RTTN, we generated RPE1-based RTTN-GFP-inducible cell lines in the RTTN^{-/-}; p53^{-/-} background, and used doxycycline to induce the expression of GFP-tagged full-length wild-type RTTN protein or various RTTN mutants."

2). In Supplementary Figure 2 I would recommend labelling the cell cycle stages above the images, or specifying what stage $t=0$ mins relates to within the figure legend.

Response: We labeled the cell cycle stages related to the time-series images presented in Supplementary Figure 2 as the reviewer suggested.

3). The authors introduce the acronym 'PCL' for the 'procentriole-like structures' that they observe in RTTN knockout cells. This acronym is already used in the centriole field for proximal centriole-like structures in insect spermatids (Blachon et al. Genetics 2009). I would recommend choosing a different definition to avoid future confusion of these terms.

Response: We used the term "primitive procentriole bodies (PPBs)" instead of "procentriole-like structures (PCLs)" in the revised manuscript.

Reviewer #2:

The manuscript by Chen et al is an exceptionally well designed and high quality study, with clear conclusions about the functional hierarchy of proteins in assembling centrioles, particularly related to the building of full-length centrioles. The authors have used multiple high-end approaches (such as CRISPR knock out cells lines, beautiful confocal and super resolution microscopy imaging, and immunoelectron microscopy) to substantiate their conclusions on centriolar assembly. The paper is well written, and is an easy read. The model at the end of the paper, and the small cartoons in some other figures help in interpreting the data effectively. The functional characterization of human

microcephaly mutations in inducible cell culture model systems would require further study in more physiologically relevant contexts using neural stem/progenitor cells, and in vivo approaches such as conditional knock out/in models (both beyond the scope of this paper). The present study is an important advance in our current understanding of how centrioles are assembled, and will be an important addition to our existing knowledge on centriole assembly after formation of cartwheels. I am happy to accept the paper as it is, without any further modifications and delay in publishing.

I have a few minor suggestions:

_The authors might consider adding statistics to some of the figures, although the data is very easily interpretable in most cases. The only figure that begs a bit of explanation is 5d, where both SPICE and Centrobilin recruitment to centrioles seem to be partially affected upon RTTN knockdown.

_The comment in discussion on dwarfism resulting from malfunctioning in cell divisions throughout the body might be carefully considered.

Response: Thank you for your thoughtful review and encouragement. We have added statistics to Figure 5 and removed the comment on dwarfism in the revised manuscript (p. 11). Our results showed that the recruitment of SPICE and Centrobilin to centrioles was not affected by RTTN depletion (Fig. 5c,d).

Reviewer #3

1). I do not agree with the conclusion that RTTN is required for centriole elongation. I do not see any evidence that a procentriole is present in the knockdown, which then does not elongate. All of the presented data suggest that RTTN plays a role early in centriole duplication. In fact, the authors state that RTTN is involved in duplication later in the manuscript.

Response: We appreciate the reviewer's comments. Our previous version of the manuscript offered the following evidence to show that RTTN is involved in centriole elongation: (1) depletion of RTTN produces shorter centrioles; (2) depletion or complete loss of RTTN does not affect the localization of early-born procentriolar proteins (STIL, SAS-6, or CPAP) to the new born centrioles, but severely affects the recruitment of later-born centriolar proteins (POC5 and POC1B) to the distal-half centrioles; and (3) depletion of RTTN can inhibit the CPAP- or CEP120-overexpression-induced formation of overly long centrioles.

Here, we provide two new evidence (Supplementary Fig. 3 and Supplementary Fig. 6) to show that RTTN does indeed play a role in centriole elongation.

Our previous version showed that complete loss of RTTN in RTTN-knockout cells could induce *de novo* procentriole-like structures (PCLs) that lack the later-born centriolar proteins, POC1B and POC5 (Fig. 1g). In the revised manuscript, we provide new evidence showing that depletion of PLK4 (a key enzyme that controls the initial steps of centriole duplication) or addition of centrinone B (a PLK4 inhibitor) could block the formation of PPBs (an update of the term “PCL”, see below) in RTTN^{-/-}; p53^{-/-} cells (Supplementary Fig. 3). Importantly, the phenotype of numerous PPBs (>4) could be effectively rescued and converted to a normal phenotype (2 or 4 centrioles/cell) by exogenous expression of wild-type RTTN-GFP (Fig. 1b). The rescued centrioles contained POC5 (a centriolar distal-portion protein, Supplementary Fig. 3e) and CEP164 (a distal appendage protein, Supplementary Fig. 3f), and exhibited the proper nine-triplet microtubule arrangement (Supplementary Fig. 3g). Collectively, our results suggest that instead of being non-functional improperly organized protein aggregates, PPBs are more likely to represent primitive procentrioles that possess the ability to form normal centrioles. We have included this result (Supplementary Fig. 3) in the revised manuscript (p4). To avoid confusion, we have changed the term “procentriole-like structures (PCLs)” to “primitive procentriole bodies (PPBs)” in the revised manuscript.

All the images presented, both light microscopy and EM, show disorganized protein masses around the mother centriole in RTTN RNAi. It seems to me that RTTN is required for properly organizing proteins early in duplication. For example, figure 5a-d show abnormal localization of all of these proteins.

Response: Our EM results revealed the presence of shorter “procentrioles” in siRTTN-treated PLK4-inducible cells (Fig. 2f). Although, these shorter procentrioles have unclear (vague) structures and are not as compact as those found in siControl cells (Fig. 2f), they possess several procentriolar proteins (SAS6, STIL, CPAP, and CEP120) (Fig. 5) previously been reported to be essential for procentriole assembly. Our superresolution microscopy and immuno-EM results suggest that RTTN colocalizes with STIL and SAS-6 (most likely encircling the SAS-6 containing cartwheel) and is encompassed by CEP295. Thus, the unclear EM images in Figure 2f probably reflect the lack of RTTN-mediated maintenance or stabilization of the highly compact procentriole structure. We have included this statement in the text of revised manuscript (p11).

The most striking is the localization of Sas6 in Figure 5 and S4b, where the WT Sas6 are clear dots, the RTTN RNAi Sas6 is a smear. Yet, the authors use a +/- readout to determine proper organization.

Response: We have added new evidence (Supplementary Fig. 6) to explain this phenomenon. As the procentriole is known to elongate during S-G2 phases, we carefully examined the elongation of procentrioles labeled with antibodies against SAS-6 and CP110 in PLK4-myc-inducible cells during S-G2 phases. The PLK4-myc-inducible cells were treated with siControl or siRTTN, exposed to aphidicolin, and then released at different time points, as described in Supplementary Fig. 6e. The cells were then immunostained with the indicated antibodies and examined by confocal

fluorescence microscopy. To estimate the procentriole length, the distance between the SAS-6 (a proximal end protein) and CP110 (a distal end protein) signals was quantified in PLK4-induced centrioles (Supplementary Fig. 6b). At the early stages of centriole elongation (0 h, early S), the newly formed CP110- and SAS-6-labeled procentrioles were observed as ring-like structures. As elongation proceeds, these ring-like structures gradually elongated (as measured by the distance between the SAS-6 and CP110 signals) to become a clear rosette pattern (15h) wherein each petal may represent an elongated centriole. A similar observation was also reported by Comartin et al. (ref. 14). In contrast, depletion of RTTN significantly blocked this procentriole elongation, often producing a smear of ring-like structures in PLK4-myc inducible cells (Supplementary Fig. 6c). Together, these findings could explain why we frequently observed a smeared ring-like structure rather than distinct dots at the early stages of centriole elongation in PLK4-myc-inducible cells. Furthermore, our findings could also explain why depletion of RTTN commonly produced a smeared ring-like structure (not a distinct petal) (Supplementary Fig. 5a-c) and shorter centrioles in PLK4-inducible cells (Fig. 2). These results strongly support our contention that RTTN contributes to centriole elongation. We have commented this effect in the revised manuscript (p5-6).

In addition, the localization of RTTN is proximal, and it is not clear how a proximal protein plays a role in elongation per se.

Response: Regarding how the proximal protein, RTTN, contributes to elongation, we hypothesize that RTTN is recruited to the inner luminal wall of newborn centrioles during early S phase, where it likely stabilizes and maintains the primitive procentrioles that contain STIL, CPAP, centrin, and the SAS-6-containing cartwheel (Fig. 1d). These RTTN-stabilized primitive procentrioles are required for the proper loading of later-born centriolar proteins (e.g., POC5 and POC1B) to the distal-half centrioles at a later stage. Consistent with this model, we found that depletion or loss of RTTN did not affect the localization of early-born centriolar proteins (SAS-6, STIL, CPAP, centrin, or CP110) to the newborn centrioles, but severely affected the recruitment of POC5 and POC1B to the distal-half centrioles. We now include this statement in the Discussion section of revised manuscript (p11).

2). The use of the Procentriole-Like must be removed. The presence of cytoplasmic protein dots should not be compared to a procentriole in any way unless there is evidence of some semblance of procentriole structure. It is not surprising that centriole proteins appear in the cytoplasm when one has removed its normal docking site. It is quite interesting that the aggregation in the cytoplasm is still controlled throughout the cell cycle, but this does not prove that RTTN controls centriole elongation.

Response: To avoid confusion, we have changed the term “procentriole-like structures (PCLs)” to “primitive procentriole bodies (PPBs)” in the revised manuscript. The PPBs are not non-functional improperly organized protein aggregates. We believe that the PPBs are more likely to represent primitive procentrioles that possess the ability to form normal centrioles. Indeed, the phenotype of

numerous PPBs (>4) could be effectively rescued and converted to a normal phenotype (2 or 4 centrioles/cell) by exogenous expression of wild-type RTTN-GFP (Fig. 1b) in rescue experiments. The rescued centrioles contained POC5 (a distal-half centriolar protein; Supplementary Fig. 3e) and CEP164 (a distal appendage protein; Supplementary Fig. 3f), and exhibited the proper nine-triplet microtubule arrangement (Supplementary Fig. 3g). Furthermore, our results show that the formation of PPBs seem to be cell-cycle regulated. Tracing with GFP-CP110 (an early-born centriolar protein) and mCherry-H2B (histone H2B, a chromosome marker; Supplementary Fig. 2a) or GFP-CP110 and mCherry-STIL (early-born centriolar proteins; Supplementary Fig. 2b) revealed that the PPBs appeared at early S phase and gradually disassembles when cells entered mitosis. Previous reports showed that STIL (ref. 8) and SAS-6 (ref. 6) are cell cycle-regulated proteins that are degraded in late mitosis. It is thus possible that the PPBs could not be formed in the absence of STIL and SAS-6 during mitosis.

REVIEWERS' COMMENTS:

Reviewer #1 (Remarks to the Author):

In their revised manuscript Chen et al. have addressed most of my major concerns and I am happy to support publication in Nature Communications.

Reviewer #2 (Remarks to the Author):

As mentioned before in my initial review, I was happy to accept the paper as in the original version. The authors have now also responded satisfactorily to my other minor comments.

Reviewer #3 (Remarks to the Author):

I have reread the previous version, the current version, and the rebuttal of the reviews. I commend the authors for addressing the issues raised by the reviewers. I believe their answers are satisfactory and that their study does support a role for RTTN in elongation. The experiments that investigate the procentrioles on the wall of the mother are quite strong, and figure S6 is a great addition.

I am still strongly skeptical of the random dots in the cytoplasm (PPBs), they are most likely protein aggregates. Even though their assembly is cell cycle dependent, that simply means that the interactions between the components are cell cycle dependent. It is the weakest part of the paper, and being in figure 1, it greatly detracts from the other stronger points. I would highly recommend burying these data in supplement and starting strong, maybe even with Figure S6 as main Figure 1 – RTTN is depleted and procentrioles don't get long.

Responses to the reviewers

We appreciate the reviewers' thoughtful comments. Our point-by-point responses to the reviewers' comments are given below.

Reviewer #1 (Remarks to the Author):

In their revised manuscript Chen et al. have addressed most of my major concerns and I am happy to support publication in Nature Communications.

Response: We thank the referee to support publication in Nature Communications.

Reviewer #2 (Remarks to the Author):

As mentioned before in my initial review, I was happy to accept the paper as in the original version. The authors have now also responded satisfactorily to my other minor comments.

Response: We thank the referee to support publication in Nature Communications.

Reviewer #3 (Remarks to the Author):

I have reread the previous version, the current version, and the rebuttal of the reviews. I commend the authors for addressing the issues raised by the reviewers. I believe their answers are satisfactory and that their study does support a role for RTTN in elongation. The experiments that investigate the procentrioles on the wall of the mother are quite strong, and figure S6 is a great addition.

I am still strongly skeptical of the random dots in the cytoplasm (PPBs), they are most likely protein aggregates. Even though their assembly is cell cycle dependent, that simply means that the interactions between the components are cell cycle dependent. It is the weakest part of the paper, and being in figure 1, it greatly detracts from the other stronger points. I would highly recommend burying these data in supplement and starting strong, maybe even with Figure S6 as main Figure 1 - RTTN is depleted and procentrioles don't get long.

Response: As reviewer 3 suggested, we reorganize the presentation flowchart of figure order in the revised manuscript. Figure 1 (previous version) now moves to the supplementary section (Fig. S3 in the revised manuscript). Fig. S6 (previous version) now moves to the main figure section (Fig. 2 in the revised manuscript). Furthermore, we move Fig. S4 (previous version) to the main figure section (Fig. 1 in the revised manuscript) for easy tracing the flow of result presentation. Finally, to reduce the reviewer's concern of PPBs as protein aggregates, we have added a sentence in the revised manuscript as follows: "However, the possibility of PPBs being as protein aggregates can't be ruled out" (p. 5 in the revised manuscript).